# Information-theoretic lower bounds for distributed statistical estimation with communication constraints

**Yuchen Zhang**[1]     **John C. Duchi**[1]     **Michael I. Jordan**[1,2]     **Martin J. Wainwright**[1,2]
[1]Department of Electrical Engineering and Computer Science and [2]Department of Statistics
University of California, Berkeley
Berkeley, CA 94720
{yuczhang,jduchi,jordan,wainwrig}@eecs.berkeley.edu

## Abstract

We establish lower bounds on minimax risks for distributed statistical estimation under a communication budget. Such lower bounds reveal the minimum amount of communication required by any procedure to achieve the centralized minimax-optimal rates for statistical estimation. We study two classes of protocols: one in which machines send messages independently, and a second allowing for interactive communication. We establish lower bounds for several problems, including various types of location models, as well as for parameter estimation in regression models.

## 1   Introduction

Rapid growth in the size and scale of datasets has fueled increasing interest in statistical estimation in distributed settings [see, e.g., 5, 23, 7, 9, 17, 2]. Modern data sets are often too large to be stored on a single machine, so that it is natural to consider methods that involve multiple machines, each assigned a smaller subset of the full dataset. An essential design parameter in such methods is the amount of communication required between machines or chips. Bandwidth limitations on network and inter-chip communication often impose significant bottlenecks on algorithmic efficiency.

The focus of the current paper is the communication complexity of various classes of statistical estimation problems. More formally, suppose that we are interested in estimating the parameter $\theta$ of some unknown distribution $P$, based on a dataset of $N$ i.i.d. samples. In the classical setting, one considers *centralized estimators* that have access to all $N$ samples, and for a given estimation problem, the optimal performance over all centralized schemes can be characterized by the minimax rate. By way of contrast, in the distributed setting, one is given $m$ different machines, and each machine is assigned a subset of samples of size $n = \lfloor \frac{N}{m} \rfloor$. Each machine may perform arbitrary operations on its own subset of data, and it then communicates results of these intermediate computations to the other processors or to a central fusion node. In this paper, we try to answer the following question: what is the minimal number of bits that must be exchanged in order to achieve the optimal estimation error achievable by centralized schemes?

There is a substantial literature on communication complexity in many settings, including function computation in theoretical computer science (e.g., [21, 1, 13]), decentralized detection and estimation (e.g., [18, 16, 15]) and information theory [11]. For instance, Luo [15] considers architectures in which machines may send only a single bit to a centralized processor; for certain problems, he shows that if each machine receives a single one-dimensional sample, it is possible to achieve the optimal centralized rate up to constant factors. Among other contributions, Balcan et al. [2] study Probably Approximately Correct (PAC) learning in the distributed setting; however, their stated lower bounds do not involve the number of machines. In contrast, our work focuses on scaling issues, both in terms of the number of machines as well as the dimensionality of the underlying data, and we formalize the problem in terms of statistical minimax theory.

More precisely, we study the following problem: given a budget $B$ of the total number of bits that may be communicated from the $m$ distributed datasets, what is the minimax risk of any estimator based on the communicated messages? While there is a rich literature connecting information-theoretic techniques with the risk of statistical estimators (e.g. [12, 22, 20, 19]), little of it characterizes the effects of limiting communication. In this paper, we present some minimax lower bounds for distributed statistical estimation. By comparing our lower bounds with results in statistical estimation, we can identify the minimal communication cost that a distributed estimator must pay to have performance comparable to classical centralized estimators. Moreover, we show how to leverage recent work [23] to achieve these fundamental limits.

## 2   Problem setting and notation

We begin with a formal description of the statistical estimation problems considered here. Let $\mathcal{P}$ denote a family of distributions and let $\theta : \mathcal{P} \to \Theta \subseteq \mathbb{R}^d$ denote a function defined on $\mathcal{P}$. A canonical example throughout the paper is the problem of mean estimation, in which $\theta(P) = \mathbb{E}_P[X]$. Suppose that, for some fixed but unknown member $P$ of $\mathcal{P}$, there are $m$ sets of data stored on individual machines, where each subset $X^{(i)}$ is an i.i.d. sample of size $n$ from the unknown distribution $P$.[1] Given this distributed collection of local data sets, our goal is to estimate $\theta(P)$ based on the $m$ samples $X^{(1)}, \ldots, X^{(m)}$, but using limited communication.

We consider a class of distributed protocols $\Pi$, in which at each round $t = 1, 2, \ldots$, machine $i$ sends a message $Y_{t,i}$ that is a measurable function of the local data $X^{(i)}$, and potentially of past messages. It is convenient to model this message as being sent to a central fusion center. Let $\overline{Y}_t = \{Y_{t,i}\}_{i \in [m]}$ denote the collection of all messages sent at round $t$. Given a total of $T$ rounds, the protocol $\Pi$ collects the sequence $(\overline{Y}_1, \ldots, \overline{Y}_T)$, and constructs an estimator $\widehat{\theta} := \widehat{\theta}(\overline{Y}_1, \ldots, \overline{Y}_T)$. The length $L_{t,i}$ of message $Y_{t,i}$ is the minimal number of bits required to encode it, and the total $L = \sum_{t=1}^{T} \sum_{i=1}^{m} L_{t,i}$ of all messages sent corresponds to the *total communication cost* of the protocol. Note that the communication cost is a random variable, since the length of the messages may depend on the data, and the protocol may introduce auxiliary randomness.

It is useful to distinguish two different classes, namely *independent* versus *interactive* protocols. An independent protocol $\Pi$ is based on a single round ($T = 1$) of communication, in which machine $i$ sends message $Y_{1,i}$ to the fusion center. Since there are no past messages, the message $Y_{1,i}$ can depend only on the local sample $X^{(i)}$. Given a family $\mathcal{P}$, the class of independent protocols with budget $B \geq 0$ is given by

$$\mathcal{A}_{\mathrm{ind}}(B, \mathcal{P}) = \left\{ \text{independent protocols } \Pi \text{ such that } \quad \sup_{P \in \mathcal{P}} \mathbb{E}_P \left[ \sum_{i=1}^{m} L_i \right] \leq B \right\}. \qquad (1)$$

(For simplicity, we use $Y_i$ to indicate the message sent from processor $i$ and $L_i$ to denote its length in the independent case.) It can be useful in some situations to have more granular control on the amount of communication, in particular by enforcing budgets on a per-machine basis. In such cases, we introduce the shorthand $B_{1:m} = (B_1, \ldots, B_m)$ and define

$$\mathcal{A}_{\mathrm{ind}}(B_{1:m}, \mathcal{P}) = \left\{ \text{independent protocols } \Pi \text{ such that } \quad \sup_{P \in \mathcal{P}} \mathbb{E}_P[L_i] \leq B_i \text{ for } i \in [m] \right\}. \qquad (2)$$

In contrast to independent protocols, the class of interactive protocols allows for interaction at different stages of the message passing process. In particular, suppose that machine $i$ sends message $Y_{t,i}$ to the fusion center at time $t$, who then posts it on a "public blackboard," where all machines can read $Y_{t,i}$. We think of this as a global broadcast system, which may be natural in settings in which processors have limited power or upstream capacity, but the centralized fusion center can send messages without limit. In the interactive setting, the message $Y_{t,i}$ should be viewed as a measurable function of the local data $X^{(i)}$, and the past messages $\overline{Y}_{1:t-1}$. The family of interactive protocols with budget $B \geq 0$ is given by

$$\mathcal{A}_{\mathrm{inter}}(B, \mathcal{P}) = \left\{ \text{interactive protocols } \Pi \text{ such that } \quad \sup_{P \in \mathcal{P}} \mathbb{E}_P[L] \leq B \right\}. \qquad (3)$$

We conclude this section by defining the minimax framework used throughout this paper. We wish to characterize the best achievable performance of estimators $\widehat{\theta}$ that are functions of only the messages $(\overline{Y}_1, \ldots, \overline{Y}_T)$. We measure the quality of a protocol and estimator $\widehat{\theta}$ by the mean-squared error

$$\mathbb{E}_{P,\Pi} \left[ \|\widehat{\theta}(\overline{Y}_1, \ldots, \overline{Y}_T) - \theta(P)\|_2^2 \right],$$

where the expectation is taken with respect to the protocol $\Pi$ and the $m$ i.i.d. samples $X^{(i)}$ of size $n$ from distribution $P$. Given a class of distributions $\mathcal{P}$, parameter $\theta : \mathcal{P} \to \Theta$, and communication budget $B$, the *minimax risk for independent protocols* is

$$\mathfrak{M}^{\mathrm{ind}}(\theta, \mathcal{P}, B) := \inf_{\Pi \in \mathcal{A}_{\mathrm{ind}}(B, \mathcal{P})} \inf_{\widehat{\theta}} \sup_{P \in \mathcal{P}} \mathbb{E}_{P,\Pi} \left[ \left\| \widehat{\theta}(Y_1, \ldots, Y_m) - \theta(P) \right\|_2^2 \right]. \tag{4}$$

Here, the infimum is taken jointly over all independent procotols $\Pi$ that satisfy the budget constraint $B$, and over all estimators $\widehat{\theta}$ that are measurable functions of the messages in the protocol. This minimax risk should also be understood to depend on both the number of machines $m$ and the individual sample size $n$. The *minimax risk for interactive protocols*, denoted by $\mathfrak{M}^{\mathrm{inter}}$, is defined analogously, where the infimum is instead taken over the class of interactive protocols. These communication-dependent minimax risks are the central objects in this paper: they provide a sharp characterization of the optimal rate of statistical estimation as a function of the communication budget $B$.

# 3 Main results

With our setup in place, we now turn to the statement of our main results, along with some discussion of their consequences.

## 3.1 Lower bound based on metric entropy

We begin with a general but relatively naive lower bound that depends only on the geometric structure of the parameter space, as captured by its metric entropy. In particular, given a subset $\Theta \subset \mathbb{R}^d$, we say $\{\theta^1, \ldots, \theta^K\}$ are $\delta$-separated if $\left\| \theta^i - \theta^j \right\|_2 \geq \delta$ for $i \neq j$. We then define the *packing entropy* of $\Theta$ as

$$\log M_\Theta(\delta) := \log_2 \left[ \max \left\{ K \in \mathbb{N} \mid \{\theta_1, \ldots, \theta^K\} \subset \Theta \text{ are } \delta\text{-separated}\right\} \right]. \tag{5}$$

The function $\theta \mapsto \log M_\Theta(\delta)$ is left-continuous and non-increasing in $\delta$, so we may define the inverse function $\log M_\Theta^{-1}(B) := \sup\{\delta \mid \log M_\Theta(\delta) \geq B\}$.

**Proposition 1** *For any family of distributions $\mathcal{P}$ and parameter set $\Theta = \theta(\mathcal{P})$, the interactive minimax risk is lower bounded as*

$$\mathfrak{M}^{\mathrm{inter}}(\theta, \mathcal{P}, B) \geq \left( \frac{1}{4} \log M_\Theta^{-1}(2B + 2) \right)^2. \tag{6}$$

Of course, the same lower bound also holds for $\mathfrak{M}^{\mathrm{ind}}(\theta, \mathcal{P}, B)$, since any independent protocol is a special case of an interactive protocol. Although Proposition 1 is a relatively generic statement, not exploiting any particular structure of the problem, it is in general unimprovable by more than constant factors, as the following example illustrates.

**Example: Bounded mean estimation.** Suppose that our goal is to estimate the mean $\theta = \theta(P)$ of a class of distributions $\mathcal{P}$ supported on the interval $[0, 1]$, so that $\Theta = \theta(\mathcal{P}) = [0, 1]$. Suppose that a single machine ($m = 1$) receives $n$ i.i.d. observations $X_i$ according to $P$. Since the packing entropy is lower bounded as $\log M_\Theta(\delta) \geq \log(1/\delta)$, the lower bound (6) implies

$$\mathfrak{M}^{\mathrm{ind}}(\theta, \mathcal{P}, B) \geq \mathfrak{M}^{\mathrm{inter}}(\theta, \mathcal{P}, B) \geq \frac{e^{-2}}{4} e^{-2B}.$$

Thus, setting $B = \frac{1}{2} \log n$ yields the lower bound $\mathfrak{M}^{\mathrm{ind}}(\theta, \mathcal{P}([0,1]), B) \geq \frac{e^{-2}}{4n}$. This lower bound is sharp up to the constant pre-factor, since it can be achieved by a simple method. Given its $n$ observations, the single machine can compute the sample mean $\overline{X}_n = \frac{1}{n} \sum_{i=1}^n X_i$. Since the sample mean lies in the interval $[0, 1]$, it can be quantized to accuracy $1/n$ using $\log(n)$ bits, and this quantized version $\widehat{\theta}$ can be transmitted. A straightforward calculation shows that $\mathbb{E}[(\widehat{\theta} - \theta)^2] \leq \frac{2}{n}$, so Proposition 1 yields an order-optimal bound in this case.

## 3.2 Multi-machine settings

We now turn to the more interesting multi-machine setting ($m > 1$). Let us study how the *budget B*—meaning the of bits required to achieve the minimax rate—scales with the number of machines $m$. We begin by considering the uniform location family $\mathcal{U} = \{P_\theta, \ \theta \in [-1, 1]\}$, where $P_\theta$ is the uniform distribution on the interval $[\theta - 1, \ \theta + 1]$. For this problem, a direct application of Proposition 1 gives a nearly sharp result.

**Corollary 1** *Consider the uniform location family $\mathcal{U}$ with $n$ i.i.d. observations per machine:*

(a) *Whenever the communication budget is upper bounded as $B \leq \log(mn)$, there is a universal constant $c$ such that*

$$\mathfrak{M}^{\text{inter}}(\theta, \mathcal{U}, B) \geq \frac{c}{(mn)^2}.$$

(b) *Conversely, given a budget of $B = \lceil 2 + 2\ln m \rceil \log(mn)$ bits, there is a universal constant $c'$ such that*

$$\mathfrak{M}^{\text{inter}}(\theta, \mathcal{U}, B) \leq \frac{c'}{(mn)^2}.$$

If each of $m$ machines receives $n$ observations, we have a total sample size of $mn$, so the minimax rate over all centralized procedures scales as $1/(mn)^2$ (for instance, see [14]). Consequently, Corollary 1(b) shows that the number of bits required to achieve the centralized rate has only *logarithmic* dependence on the number $m$ of machines. Part (a) shows that this logarithmic dependence on $m$ is unavoidable.

It is natural to wonder whether such logarithmic dependence holds more generally. The following result shows that it does not: for some problems, the dependence on $m$ must be (nearly) linear. In particular, we consider estimation in a normal location family model, where each machine receives an i.i.d. sample of size $n$ from a normal distribution $\mathsf{N}(\theta, \sigma^2)$ with unknown mean $\theta$.

**Theorem 1** *For the univariate normal family $\mathcal{N} = \{\mathsf{N}(\theta, \sigma^2) \mid \theta \in [-1, 1]\}$, there is a universal constant $c$ such that*

$$\mathfrak{M}^{\text{inter}}(\theta, \mathcal{N}, B) \geq c \, \frac{\sigma^2}{mn} \, \min\left\{ \frac{mn}{\sigma^2}, \ \frac{m}{\log m}, \ \frac{m}{B \, \log m} \right\}. \tag{7}$$

The centralized minimax rate for estimating a univariate normal mean based on $mn$ observations is $\frac{\sigma^2}{mn}$; consequently, the lower bound (7) shows that at least $B = \Omega\left(\frac{m}{\log m}\right)$ bits are required for a decentralized procedure to match the centralized rate in this case. This type of scaling is dramatically different than the logarithmic scaling for the uniform family, showing that establishing sharp communication-based lower bounds requires careful study of the underlying family of distributions.

## 3.3 Independent protocols in multi-machine settings

Departing from the interactive setting, in this section we focus on independent protocols, providing somewhat more general results than those for interactive protocols. We first provide lower bounds for the problem of mean estimation in the parameter for a $d$-*dimensional normal location family*

$$\mathcal{N}_d = \{\mathsf{N}(\theta, \sigma^2 I_{d \times d}) \mid \theta \in \Theta = [-1, 1]^d\}, \tag{8}$$

**Theorem 2** *For $i = 1, \ldots, m$, assume that each machine has communication budget $B_i$, and receives an i.i.d. sample of size $n$ from a distribution $P \in \mathcal{N}_d$. There exists a universal (numerical) constant $c$ such that*

$$\mathfrak{M}^{\text{ind}}(\theta, \mathcal{N}_d, B_{1:m}) \geq c \frac{\sigma^2 d}{mn} \, \min\left\{ \frac{mn}{\sigma^2}, \ \frac{m}{\log m}, \ \frac{m}{\left(\sum_{i=1}^m \min\{1, \frac{B_i}{d}\}\right) \, \log m} \right\}. \tag{9}$$

Given centralized access to the full $mn$-sized sample, a reasonable procedure would be to compute the sample mean, leading to an estimate with mean-squared error $\frac{\sigma^2 d}{mn}$, which is minimax optimal.

Consequently, Theorem 2 shows that to achieve an order-optimal mean-squared error, the total number of bits communicated must (nearly) scale with the product of the dimension $d$ and number of machines $m$, that is, as $dm/\log m$. If we ignore logarithmic factors, this lower bound is achievable by a simple procedure: each machine computes the sample mean of its local data and quantizes each coordinate to precision $\sigma^2/n$ using $\mathcal{O}(d\log(n/\sigma^2))$ bits. These quantized sample averages are communicated to the fusion center using $B = \mathcal{O}(dm\log(n/\sigma^2))$ total bits. The fusion center averages them, obtaining an estimate with mean-squared error of optimal order $\sigma^2 d/(mn)$ as required.

We finish this section by presenting a result that is sharp up to numerical constant prefactors. It is a minimax lower bound for mean estimation over the family $\mathcal{P}_d = \{P \text{ supported on } [-1,1]^d\}$.

**Proposition 2** *Assume that each of $m$ machines receives a single sample ($n = 1$) from a distribution in $\mathcal{P}_d$. There exists a universal (numerical) constant $c$ such that*

$$\mathfrak{M}^{\mathrm{ind}}(\theta, \mathcal{P}_d, B_{1:m}) \geq c \frac{d}{m} \min\left\{m, \frac{m}{\sum_{i=1}^m \min\{1, \frac{B_i}{d}\}}\right\}, \tag{10}$$

*where $B_i$ is the budget for machine $i$.*

The standard minimax rate for $d$-dimensional mean estimation scales as $d/m$. The lower bound (10) shows that in order to achieve this scaling, we must have $\sum_{i=1}^m \min\{1, \frac{B_i}{d}\} \gtrsim m$, showing that each machine must send $B_i \gtrsim d$ bits.

Moreover, this lower bound is achievable by a simple scheme. Suppose that machine $i$ receives a $d$-dimensional vector $X_i \in [-1,1]^d$. Based on $X_i$, it generates a Bernoulli random vector $Z_i = (Z_{i1}, \ldots, Z_{id})$ with $Z_{ij} \in \{0,1\}$ taking the value 1 with probability $(1 + X_{ij})/2$, independently across coordinates. Machine $i$ uses $d$ bits to send the vector $Z_i \in \{0,1\}^d$ to the fusion center. The fusion center then computes the average $\hat{\theta} = \frac{1}{m}\sum_{i=1}^m (2Z_i - 1)$. This average is unbiased, and its expected squared error is bounded by $d/m$.

## 4 Consequences for regression

In this section, we turn to identifying the minimax rates for a pair of important estimation problems: linear regression and probit regression.

### 4.1 Linear regression

We consider a distributed instantiation of linear regression with fixed design matrices. Concretely, suppose that each of $m$ machines has stored a fixed design matrix $A^{(i)} \in \mathbb{R}^{n \times d}$ and then observes a response vector $b^{(i)} \in \mathbb{R}^d$ from the standard linear regression model

$$b^{(i)} = A^{(i)}\theta + \varepsilon^{(i)}, \tag{11}$$

where $\varepsilon^{(i)} \sim \mathsf{N}(0, \sigma^2 I_{n \times n})$ is a noise vector. Our goal is to estimate unknown regression vector $\theta \in \Theta = [-1,1]^d$, shared across all machines, in a distributed manner, To state our result, we assume uniform upper and lower bounds on the eigenvalues of the rescaled design matrices, namely

$$0 < \lambda_{\min} \leq \min_{i \in \{1,\ldots,m\}} \frac{\eta_{\min}(A^{(i)})}{\sqrt{n}} \quad \text{and} \quad \max_{i \in \{1,\ldots,m\}} \frac{\eta_{\max}(A^{(i)})}{\sqrt{n}} \leq \lambda_{\max}. \tag{12}$$

**Corollary 2** *Consider an instance of the linear regression model (11) under condition (12).*

*(a) Then there is a universal positive constant $c$ such that*

$$\mathfrak{M}^{\mathrm{ind}}(\theta, \mathcal{P}, B_{1:m}) \geq c \frac{\sigma^2 d}{mn} \min\left\{\frac{mn}{\sigma^2}, \frac{m}{\lambda_{\max}^2 \log m}, \frac{m}{\lambda_{\max}^2 \left(\sum_{i=1}^m \min\{1, \frac{B_i}{d}\}\right)\log m}\right\}.$$

*(b) Conversely, given budgets $B_i \geq d\log(mn)$ for $i = 1, \ldots, m$, there is a universal constant $c'$ such that*

$$\mathfrak{M}^{\mathrm{ind}}(\theta, \mathcal{P}, B_{1:m}) \leq \frac{c'}{\lambda_{\min}^2} \frac{\sigma^2 d}{mn}.$$

It is a classical fact (e.g. [14]) that the minimax rate for $d$-dimensional linear regression scales as $d\sigma^2/(nm)$. Part (a) of Corollary 2 shows this optimal rate is attainable only if the budget $B_i$ at each machine is of the order $d/\log(m)$, meaning that the total budget $B = \sum_{i=1}^{m} B_i$ must grow as $\frac{dm}{\log m}$. Part (b) of the corollary shows that the minimax rate is achievable with budgets that match the lower bound up to logarithmic factors.

**Proof:** The proof of part (b) follows from techniques of Zhang et al. [23], who show that solving each regression problem separately and then performing a form of approximate averaging, in which each machine uses $B_i = d\log(mn)$ bits, achieves the minimax rate up to constant prefactors.

To prove part (a), we show that solving an arbitrary Gaussian mean estimation problem can be reduced to solving a specially constructed linear regression problem. This reduction allows us to apply the lower bound from Theorem 2. Given $\theta \in \Theta$, consider the Gaussian mean model

$$X^{(i)} = \theta + w^{(i)}, \quad \text{where } w^{(i)} \sim \mathsf{N}\left(0, \frac{\sigma^2}{\lambda_{\max}^2 n} I_{d \times d}\right).$$

Each machine $i$ has its own design matrix $A^{(i)}$, and we use it to construct a response vector $b^{(i)} \in \mathbb{R}^n$. Since $\eta_{\max}(A^{(i)}/\sqrt{n}) \leq \lambda_{\max}$, the matrix $\Sigma^{(i)} := \sigma^2 I_{n \times n} - \frac{\sigma^2}{\lambda_{\max}^2 n} A^{(i)}(A^{(i)})^\top$ is positive semidefinite. Consequently, we may form a response vector via

$$b^{(i)} = A^{(i)} X^{(i)} + z^{(i)}, \quad z^{(i)} \sim \mathsf{N}\left(0, \Sigma^{(i)}\right) \text{ is drawn independently of } w^{(i)}. \tag{13}$$

The independence of $w^{(i)}$ and $z^{(i)}$ guarantees that $b^{(i)} \sim \mathsf{N}(A^{(i)}\theta, \sigma^2 I_{n \times n})$, so that the pair $(b^{(i)}, A^{(i)})$ is faithful to the regression model (11).

Now consider any protocol $\Pi \in \mathcal{A}_{\mathrm{ind}}(B, \mathcal{P})$ that can solve any regression problem to within accuracy $\delta$, so that $\mathbb{E}[\|\widehat{\theta} - \theta\|_2^2] \leq \delta^2$. By the previously described reduction, the protocol $\Pi$ can also solve the mean estimation problem to accuracy $\delta$, in particular via the pair $(A^{(i)}, b^{(i)})$ constructed via expression (13). Combined with this reduction, the corollary thus follows from Theorem 2. ∎

### 4.2 Probit regression

We now turn to the problem of binary classification, in particular considering the probit regression model. As in the previous section, each of $m$ machines has a fixed design matrix $A^{(i)} \in \mathbb{R}^{n \times d}$, where $A^{(i,k)}$ denotes the $k$th row of $A^{(i)}$. Machine $i$ receives $n$ binary responses $Z^{(i)} = (Z^{(i,1)}, \ldots, Z^{(i,n)})$, drawn from the conditional distribution

$$\mathbb{P}(Z^{(i,k)} = 1 \mid A^{(i,k)}, \theta) = \Phi(A^{(i,k)}\theta) \quad \text{for some fixed } \theta \in \Theta = [-1, 1]^d, \tag{14}$$

where $\Phi(\cdot)$ denotes the standard normal CDF. The log-likelihood of the probit model (14) is concave [4, Exercise 3.54]. Under condition (12) on the design matrices, we have:

**Corollary 3** *Consider the probit model* (14) *under condition* (12). *Then*

(a) *There is a universal constant $c$ such that*

$$\mathfrak{M}^{\mathrm{ind}}(\theta, \mathcal{P}, B_{1:m}) \geq c\frac{d}{mn} \min\left\{mn, \frac{m}{\lambda_{\max}^2 \log m}, \frac{m}{\lambda_{\max}^2 \left(\sum_{i=1}^{m} \min\{1, \frac{B_i}{d}\}\right) \log m}\right\}.$$

(b) *Conversely, given budgets $B_i \geq d\log(mn)$ for $i = 1, \ldots, m$, there is a universal constant $c'$ such that*

$$\mathfrak{M}^{\mathrm{ind}}(\theta, \mathcal{P}, B_{1:m}) \leq \frac{c'}{\lambda_{\min}^2} \frac{d}{mn}.$$

**Proof:** As in the previous case with linear regression, Zhang et al.'s study of distributed convex optimization [23] gives part (b): each machine solves the local probit regression separately, after which each machine sends $B_i = d\log(mn)$ bits to average its local solution.

To prove part (a), we show that linear regression problems can be solved via estimation in a specially constructed probit model. Consider an arbitrary $\theta \in \Theta$; assume we have a regression problem of the

form (11) with noise variance $\sigma^2 = 1$. We construct the binary responses for our probit regression $(Z^{(i,1)}, \ldots, Z^{(i,n)})$ by

$$Z^{(i,k)} = \begin{cases} 1 & \text{if } b^{(i,k)} \geq 0, \\ 0 & \text{otherwise.} \end{cases} \tag{15}$$

By construction, we have $\mathbb{P}(Z^{(i,k)} = 1 \mid A^{(i)}, \theta) = \Phi(A^{(i,k)}\theta)$ as desired for our model (14). By inspection, any protocol $\Pi \in \mathcal{A}_{\text{ind}}(B, \mathcal{P})$ solving the probit regression problem provides an estimator with the same error (risk) as the original linear regression problem via the construction (15). Corollary 2 provides the desired lower bound. ∎

## 5 Proof sketches for main results

We now give an outline of the proof of each of our main results (Theorems 1 and 2), providing a more detailed proof sketch for Proposition 2, since it displays techniques common to our arguments.

### 5.1 Broad outline

Most of our lower bounds follow the same basic strategy of reducing an estimation problem to a testing problem. Following this reduction, we then develop inequalities relating the probability of error in the test to the number of bits contained in the messages $Y_i$ sent from each machine. Establishing these links is the most technically challenging aspect.

Our reduction from estimation to testing is somewhat more general than the classical reductions (e.g., [22, 20]), since we do not map the original estimation problem to a strict test, but rather a test that allows some errors. Let $\mathcal{V}$ denote an index set of finite cardinality, where $\nu \in \mathcal{V}$ indexes a family of probability distributions $\{P(\cdot \mid \nu)\}_{\nu \in \mathcal{V}}$. For each member of this family, associate with a parameter $\theta_\nu := \theta(P(\cdot \mid \nu)) \in \Theta$, where $\Theta$ denotes the parameter space. In our proofs applicable to $d$-dimensional problems, we set $\mathcal{V} = \{-1, 1\}^d$, and we index vectors $\theta_\nu$ by $\nu \in \mathcal{V}$. Now, we sample $V$ uniformly at random from $\mathcal{V}$. Conditional on $V = \nu$, we then sample $X$ from a distribution $P_X(\cdot \mid V = \nu)$ satisfying $\theta_\nu := \theta(P_X(\cdot \mid \nu)) = \delta\nu$, where $\delta > 0$ is a fixed quantity that we control. We define $d_{\text{ham}}(\nu, \nu')$ to be the Hamming distance between $\nu, \nu' \in \mathcal{V}$. This construction gives

$$\|\theta_\nu - \theta_{\nu'}\|_2 = 2\delta \sqrt{d_{\text{ham}}(\nu, \nu')}.$$

Fixing $t \in \mathbb{R}$, the following lemma reduces the problem of estimating $\theta$ to finding a point $\nu \in \mathcal{V}$ within distance $t$ of the random variable $V$. The result extends a result of Duchi and Wainwright [8]; for completeness we provide a proof in Appendix H.

**Lemma 1** *Let $V$ be uniformly sampled from $\mathcal{V}$. For any estimator $\widehat{\theta}$ and any $t \in \mathbb{R}$, we have*

$$\sup_{P \in \mathcal{P}} \mathbb{E}[\|\widehat{\theta} - \theta(P)\|_2^2] \geq \delta^2(\lfloor t \rfloor + 1) \inf_{\widehat{\nu}} \mathbb{P}\left(d_{\text{ham}}(\widehat{\nu}, V) > t\right),$$

*where the infimum ranges over all testing functions.*

Lemma 1 shows that minimax lower lower bound can be derived by showing that, for some $t > 0$ to be chosen, it is difficult to identify $V$ within a radius of $t$. The following extension of Fano's inequality [8] can be used to control this type of error probability:

**Lemma 2** *Let $V \to X \to \widehat{V}$ be a Markov chain, where $V$ is uniform on $\mathcal{V}$. For any $t \in \mathbb{R}$, we have*

$$\mathbb{P}(d_{\text{ham}}(\widehat{V}, V) > t) \geq 1 - \frac{I(V; X) + \log 2}{\log \frac{|\mathcal{V}|}{N_t}},$$

*where $N_t := \max_{\nu \in \mathcal{V}} |\{\nu' \in \mathcal{V} : d_{\text{ham}}(\nu, \nu') \leq t\}|$ is the size of the largest $t$-neighborhood in $\mathcal{V}$.*

Lemma 2 allows flexibility in the application of the minimax bounds from Lemma 1. If there is a large set $\mathcal{V}$ for which it is easy to control $I(V; X)$, whereas neighborhoods in $\mathcal{V}$ are relatively small (i.e., $N_t$ is small), then we can obtain sharp lower bounds.

In a distributed protocol, we have a Markov chain $V \to X \to Y$, where $Y$ denotes the messages the different machines send. Based on the messages $Y$, we consider an arbitrary estimator $\widehat{\theta}(Y)$. For $0 \le t \le \lceil d/3 \rceil$, we have $N_t = \sum_{\tau=0}^{t} \binom{d}{\tau} \le 2\binom{d}{t}$. Since $\binom{d}{t} \le (de/t)^t$, for $t \le d/6$ we have

$$\log \frac{|\mathcal{V}|}{N_t} \ge d \log 2 - \log 2 \binom{d}{t} \ge d \log 2 - \frac{d}{6} \log(6e) - \log 2 = d \log \frac{2}{2^{1/d} \sqrt[6]{6e}} > \frac{d}{6}$$

for $d \ge 12$ (the case $d < 12$ can be checked directly). Thus, combining Lemma 1 and Lemma 2 (using the Markov chain $V \to X \to Y \to \widehat{\theta}$), we find that for $t = \lfloor d/6 \rfloor$,

$$\sup_{P \in \mathcal{P}} \mathbb{E}\left[ \|\widehat{\theta}(Y) - \theta(P)\|_2^2 \right] \ge \delta^2(\lfloor d/6 \rfloor + 1)\left( 1 - \frac{I(Y;V) + \log 2}{d/6} \right). \tag{16}$$

With inequality (16) in hand, it then remains to upper bound the mutual information $I(Y;V)$, which is the main technical content of each of our results.

## 5.2 Proof sketch of Proposition 2

Following the general outline of the previous section, let $V$ be uniform on $\mathcal{V} = \{-1,1\}^d$. Letting $0 < \delta \le 1$ be a positive number, for $i \in [m]$ we independently sample $X^{(i)} \in \mathbb{R}^d$ according to

$$P(X_j^{(i)} = \nu_j \mid V = \nu) = \frac{1+\delta}{2} \quad \text{and} \quad P(X_j^{(i)} = -\nu_j \mid V = \nu) = \frac{1-\delta}{2}. \tag{17}$$

Under this distribution, we can give a sharp characterization of the mutual information $I(V;Y_i)$. In particular, we show in Appendix B that under the sampling distribution (17), there exists a numerical constant $c$ such that

$$I(V;Y_i) \le c\delta^2 I(X^{(i)};Y_i). \tag{18}$$

Since the random variable $X$ takes discrete values, we have

$$I(X^{(i)};Y_i) \le \min\{H(X^{(i)}), H(Y_i)\} \le \min\{d, H(Y_i)\}.$$

Since the expected length of message $Y_i$ is bounded by $B_i$, Shannon's source coding theorem [6] implies that $H(Y_i) \le B_i$. In particular, inequality (18) establishes a link between the initial distribution (17) and the number of bits used to transmit information, that is,

$$I(V;Y_i) \le c\delta^2 \min\{d, B_i\}. \tag{19}$$

We can now apply the quantitative data processing inequality (19) in the bound (16). By the independence of the communication scheme, $I(V;Y_{1:m}) \le \sum_{i=1}^{m} I(V;Y_i)$, and thus inequality (16) simplifies to

$$\mathfrak{M}^{\mathrm{ind}}(\theta, \mathcal{P}, B_{1:m}) \ge \delta^2(\lfloor d/6 \rfloor + 1)\left( 1 - \frac{c\delta^2 \sum_{i=1}^{m} \min\{d, B_i\} + \log 2}{d/6} \right).$$

Assuming $d \ge 9$, so $1 - 6\log 2/d > 1/2$, we see that choosing $\delta^2 = \min\{1, \frac{d}{24c \sum_{i=1}^{m} \min\{B_i, d\}}\}$ implies

$$\mathfrak{M}^{\mathrm{ind}}(\theta, \mathcal{P}, B_{1:m}) \ge \frac{\delta^2(\lfloor d/6 \rfloor + 1)}{4} = \frac{\lfloor d/6 \rfloor + 1}{4} \min\left\{ 1, \frac{d}{24c \sum_{i=1}^{m} \min\{B_i, d\}} \right\}.$$

Rearranging slightly gives the statement of the proposition.

## Acknowledgments

We thank the anonymous reviewers for their helpful feedback and comments. JCD was supported by a Facebook Graduate Fellowship. Our work was supported in part by the U.S. Army Research Laboratory, U.S. Army Research Office under grant number W911NF-11-1-0391, and Office of Naval Research MURI grant N00014-11-1-0688.

## Footnotes

[1] Although we assume in this paper that every machine has the same amount of data, our technique generalizes easily to prove tight lower bounds for distinct data sizes on different machines.

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
