[Supplementary Material]

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

# Appendices

## A  Notation and proof setup

In these appendices, we provide the proofs of all our major results. Note that we prove the theorems out of the order in which they are presented: many of the theorems build on one another, so we present them in (rough) order of most basic to most complex. Before proceeding to the proofs proper, we give notation.

**Notation in proofs**

The distributed machines are indexed by $i \in \{1, \ldots, m\}$. For machine $i$, it receives local dataset $D_i$. If $D_i$ contains multiple examples, we may denote the $k$-th example by $X^{(i,k)}$. If each example has more than one coordinate, then the $j$-th coordinate is represented by $X_j^{(i,k)}$.

For a random variable $X$, we let $P_X$ denote the probability measure on $X$, so that $P_X(S) = P(X \in S)$, and we abuse notation by writing $p_X$ for the probability mass function or density of $X$, depending on the situation, so that $p_X(x) = P(X = x)$ in the discrete case and denotes the density of $X$ at $x$ when $p_X$ is a density. For discrete random variable $X$, we let $H(X) = -\sum_x p_X(x) \log p_X(x)$ denote the (Shannon) entropy, and for probability distributions $P, Q$ on a set $\mathcal{X}$, with densities $p, q$ with respect to a base measure $\mu$, we write the KL-divergence as

$$D_{\mathrm{kl}}\left(P \| Q\right) := \int_{\mathcal{X}} p(x) \log \frac{p(x)}{q(x)} d\mu(x).$$

The mutual information $I(X; Y)$ between random variables $X$ and $Y$ where $Y$ has distribution $P_Y$ is defined as

$$I(X; Y) := \mathbb{E}_{P_X}\left[D_{\mathrm{kl}}\left(P_Y(\cdot \mid X) \| P_Y(\cdot)\right)\right] = \int D_{\mathrm{kl}}\left(P_Y(\cdot \mid X = x) \| P_Y(\cdot)\right) dP_X(x).$$

**Le Cam's method**

In low-dimensional settings, it is sometimes difficult to apply our incarnation of Fano's inequality as outlined in Section 5.1. In these settings, we use a minimax lower bound based on a two-point family. In this setting, we let $\mathcal{V} = \{-1, 1\}$, and define $\theta_\nu = \theta(P_\nu)$ as usual. Then Le Cam's inequality (e.g. [22] or [19, Theorem 2.2]) guarantees that for $V$ chosen uniformly as $V = 1$ or $V = -1$ we have

$$\inf_{\widehat{\nu}} \mathbb{P}(\widehat{\nu} \neq V) \geq \frac{1}{2} - \frac{1}{2} \|P_1 - P_{-1}\|_{\mathrm{TV}}.$$

As a consequence, if by construction $\theta_\nu = \delta\nu$, then Lemma 1 implies that

$$\inf_{\widehat{\theta}} \max_{P \in \{P_1, P_{-1}\}} \mathbb{E}[\|\widehat{\theta} - \theta(P)\|_2^2] \geq \delta^2 \left(\frac{1}{2} - \frac{1}{2} \|P_1 - P_{-1}\|_{\mathrm{TV}}\right). \tag{20}$$

We use arguments based on Le Cam's method (20) when the dimension $d$ is small.

In addition, it will be useful to have a few simple upper bounds on the distance $\|P_1 - P_{-1}\|_{\mathrm{TV}}$. We claim that if we have the Markov chain $V \to Y$, for any random variable $Y$, then for $V$ chosen uniformly in a set $\mathcal{V} = \{\nu, \nu'\}$,

$$\|P_Y(\cdot \mid V = \nu) - P_Y(\cdot \mid V = \nu')\|_{\mathrm{TV}}^2 \leq 2I(Y, V). \tag{21}$$

To see inequality (21), let $P_\nu$ be shorthand for $P_Y(\cdot \mid V = \nu)$. The triangle inequality implies that

$$\|P_\nu - P_{\nu'}\|_{\mathrm{TV}} \leq \|P_\nu - (1/2)(P_\nu + P_{\nu'})\|_{\mathrm{TV}} + \frac{1}{2} \|P_\nu - P_{\nu'}\|_{\mathrm{TV}},$$

and similarly swapping the roles of $\nu'$ and $\nu$, whence

$$\|P_\nu - P_{\nu'}\|_{\mathrm{TV}} \leq 2 \min\{\|P_\nu - (1/2)(P_{\nu'} + P_\nu)\|_{\mathrm{TV}}, \|P_{\nu'} - (1/2)(P_{\nu'} + P_\nu)\|_{\mathrm{TV}}\}.$$

By Pinsker's inequality, we thus have the upper bound

$$\|P_\nu - P_{\nu'}\|_{\mathrm{TV}}^2 \leq 2 \min\{D_{\mathrm{kl}}\left(P_\nu \| (1/2)(P_\nu + P_{\nu'})\right), D_{\mathrm{kl}}\left(P_{\nu'} \| (1/2)(P_\nu + P_{\nu'})\right)\}$$

$$\leq D_{\mathrm{kl}}\left(P_\nu \| (1/2)(P_\nu + P_{\nu'})\right) + D_{\mathrm{kl}}\left(P_{\nu'} \| (1/2)(P_\nu + P_{\nu'})\right) = 2I(Y; V)$$

by the definition of mutual information.

**Tensorization of information**

We also require a type of tensorization inequality in each of our proofs for independent protocols. When $Y_i$ is constructed based only on $X^{(i)}$, we have

$$I(V; Y_{1:m}) = \sum_{i=1}^{m} I(V; Y_i \mid Y_{1:i-1}) = \sum_{i=1}^{m} H(Y_i \mid Y_{1:i-1}) - H(Y_i \mid V, Y_{1:i-1})$$

$$\leq \sum_{i=1}^{m} H(Y_i) - H(Y_i \mid V, Y_{1:i-1})$$

$$= \sum_{i=1}^{m} H(Y_i) - H(Y_i \mid V) = \sum_{i=1}^{m} I(V; Y_i) \qquad (22)$$

where we have used that conditioning reduces entropy and $Y_i$ is conditionally independent of $Y_{1:i-1}$ given $V$.

## B  Proof of Proposition 2

The proof of this proposition follows the basic outline described in Section 5.

We first describe the distribution of the step $V \to X$. Given $\nu \in \mathcal{V}$, we assume that each machine $i$ receives a $d$-dimensional sample $X^{(i)}$ with coordinates independently sampled according to

$$P(X_j = \nu_j \mid \nu) = \frac{1 + \delta\nu_j}{2} \quad \text{and} \quad P(X_j = -\nu_j \mid \nu) = \frac{1 - \delta\nu_j}{2}.$$

Let $\delta \leq \frac{1}{4}$. Then $\theta_\nu = \mathbb{E}_\nu[X]$, and moreover we have the likelihood ratio bound

$$\frac{P(X_j \in S \mid \nu)}{P(X_j \in S \mid \nu')} \leq \frac{1 + \delta}{1 - \delta} \leq \exp\left(\frac{17}{8}\delta\right), \quad \text{and} \quad \exp\left(\frac{17}{4}\delta\right) \leq 1 + 8\delta.$$

We now present a lemma that relates this ratio bound via a type of quantitative data processing inequality. The lemma is actually somewhat more general than what we require, and we prove it in Section B.1. The result is similar to recent results of Duchi et al. [10, Theorems 1 and 2], who show similar strong data processing inequalities in the context of privacy-preserving data analysis. Our proof, however, is different, as we have the Markov chain $V \to X \to Y$, and instead of a likelihood ratio bound on the channel $X \to Y$, we place a likelihood ratio bound on $V \to X$.

**Lemma 3** *Let $V$ be sampled uniformly at random from $\{-1, 1\}^d$. For any $(i, j)$, assume that $X_j^{(i)}$ is independent of $\{X_{j'}^{(i)} : j' \neq j\} \cup \{V_{j'} : j' \neq j\}$ given $V_j$. Let $\mathbb{P}_{X_j}$ be the probability measure of $X_j^{(i)}$ and assume in addition that*

$$\sup_{S \in \sigma(X_j)} \frac{\mathbb{P}_{X_j}(S \mid V = \nu)}{\mathbb{P}_{X_j}(S \mid V = \nu')} \leq \exp(\alpha).$$

*Then*

$$I(V; Y_i) \leq 2(e^{2\alpha} - 1)^2 I(X^{(i)}; Y_i).$$

Lemma 3 provides a quantitative data processing inequality relating the mutual information in the channel $X^{(i)} \to Y_i$ to that in $V \to Y_i$. In particular, we find that

$$I(V; Y_i) \leq 2\left(e^{(17/4)\delta} - 1\right)^2 I(X^{(i)}; Y_i) \leq 128\delta^2 I(X^{(i)}; Y_i).$$

This is the claimed strong data processing inequality (18), which almost completes our proof. To complete the proof, note that $\theta(P(\cdot \mid V = \nu)) = \mathbb{E}[Y \mid \nu] = \delta\nu$. Recalling the tensorization inequality (22), we also have

$$I(V; Y_{1:m}) \leq \sum_{i=1}^{m} I(V; Y_i) \leq 128\delta^2 \sum_{i=1}^{m} I(Y_i; X^{(i)}).$$

The remainder of the proof we break into two cases: when $d \geq 9$ and when $d < 9$. For the case $d \geq 9$, our proof sketch in Section 5.2, beginning from inequality (18) with $c = 128$, completes the proof. When $d < 9$, we use a slightly different argument. By a reduction to smaller dimensions, we may assume without loss of generality that $d = 1$, and we set $\mathcal{V} = \{-1, 1\}$. In this case, Le Cam's method (20) coupled with the subsequent information inequality (21) implies that

$$\mathfrak{M}^{\mathrm{ind}}(\theta, \mathcal{P}, B_{1:m}) \geq \delta^2 \left( \frac{1}{2} - \frac{1}{2} \sqrt{2 I(V; Y_{1:m})} \right). \tag{23}$$

Applying our previous bound $I(V; Y_{1:m}) \leq 128\delta^2 \sum_{i=1}^m I(Y_i; X^{(i)})$, and noting that $I(X^{(i)}; Y_i) \leq \min\{H(X^{(i)}), H(Y_i)\} \leq \min\{1, H(Y_i)\}$ since $X^{(i)} \in \{-1, 1\}$, we obtain

$$\mathfrak{M}^{\mathrm{ind}}(\theta, \mathcal{P}, B_{1:m}) \geq \delta^2 \left( \frac{1}{2} - 8 \left( \delta^2 \sum_{i=1}^m \min\{1, H(Y_i)\} \right)^{\frac{1}{2}} \right).$$

Since $H(Y_i) \leq B_i$ by Shannon's source coding theorem [6], setting

$$\delta^2 = \min \left\{ 1, \frac{1}{400 \sum_{i=1}^m \min\{1, B_i\}} \right\}$$

completes the proof.

## B.1 Proof of Lemma 3

Let $Y = Y_i$; we suppress the dependence on the index $i$ (and similarly let $X = X^{(i)}$ denote a single fixed sample). We begin with the simple observation that, by the chain rule for mutual information,

$$I(V; Y) = \sum_{j=1}^d I(V_j; Y \mid V_{1:j-1}).$$

Using the definition of mutual information and non-negativity of the KL-divergence, we have

$$I(V_j; Y \mid V_{1:j-1}) = \mathbb{E}_{V_{1:j-1}} \left[ \mathbb{E}_Y \left[ D_{\mathrm{kl}} \left( P_{V_j}(\cdot \mid Y, V_{1:j-1}) \| P_{V_j}(\cdot \mid V_{1:j-1}) \right) \mid V_{1:j-1} \right] \right]$$
$$\leq \mathbb{E}_{V_{1:j-1}} \left[ \mathbb{E}_Y \left[ D_{\mathrm{kl}} \left( P_{V_j}(\cdot \mid Y, V_{1:j-1}) \| P_{V_j}(\cdot \mid V_{1:j-1}) \right) \right.\right.$$
$$\left.\left. + D_{\mathrm{kl}} \left( P_{V_j}(\cdot \mid V_{1:j-1}) \| P_{V_j}(\cdot \mid Y, V_{1:j-1}) \right) \mid V_{1:j-1} \right] \right].$$

Now, we require an argument that builds off of a technical lemma we present in Appendix G, Lemma 8. We claim that Lemma 8 implies that

$$|P(V_j = \nu_j \mid V_{1:j-1}, Y) - P(V_j = \nu_j \mid V_{1:j-1})|$$
$$\leq 2(e^{2\alpha} - 1) \min \{ P(V_j = \nu_j \mid V_{1:j-1}, Y), P(V_j = \nu_j \mid V_{1:j-1}) \}$$
$$\times \left\| P_{X_j}(\cdot \mid V_{1:j-1}, Y) - P_{X_j}(\cdot \mid V_{1:j-1}) \right\|_{\mathrm{TV}}. \tag{24}$$

Indeed, making the identification

$$V_j \leftrightarrow A, \quad X_j \leftrightarrow B, \quad V_{1:j-1} \leftrightarrow C, \quad Y \leftrightarrow D$$

gives inequality (24), by our independence assumptions. Expanding our KL divergence bound, we have

$$D_{\mathrm{kl}} \left( P_{V_j}(\cdot \mid Y, V_{1:j-1}) \| P_{V_j}(\cdot \mid V_{1:j-1}) \right)$$
$$\leq \sum_{\nu_j} \left( P_{V_j}(\nu_j \mid Y, V_{1:j-1}) - P_{V_j}(\nu_j \mid V_{1:j-1}) \right) \log \frac{P_{V_j}(\nu_j \mid Y, V_{1:j-1})}{P_{V_j}(\nu_j \mid V_{1:j-1})}.$$

Now, using the elementary inequality for $a, b \geq 0$ that

$$\left| \log \frac{a}{b} \right| \leq \frac{|a - b|}{\min\{a, b\}},$$

we have

$$\left(P_{V_j}(\nu_j \mid Y, V_{1:j-1}) - P_{V_j}(\nu_j \mid V_{1:j-1})\right) \log \frac{P_{V_j}(\nu_j \mid Y, V_{1:j-1})}{P_{V_j}(\nu_j \mid V_{1:j-1})}$$

$$\leq \frac{(P_{V_j}(\nu_j \mid Y, V_{1:j-1}) - P_{V_j}(\nu_j \mid V_{1:j-1}))^2}{\min\{P_{V_j}(\nu_j \mid Y, V_{1:j-1}), P_{V_j}(\nu_j \mid V_{1:j-1})\}}$$

$$\leq 4(e^{2\alpha} - 1)^2 \min\left\{P_{V_j}(\nu_j \mid Y, V_{1:j-1}), P_{V_j}(\nu_j \mid V_{1:j-1})\right\}$$

$$\times \left\|P_{X_j}(\cdot \mid V_{1:j-1}, Y) - P_{X_j}(\cdot \mid V_{1:j-1})\right\|_{\mathrm{TV}}^2$$

by inequality (24).

Substituting this into our bound on KL-divergence, we obtain

$$I(V_j; Y \mid V_{1:j-1})$$

$$= \mathbb{E}_{V_{1:j-1}} \left[\mathbb{E}_Y \left[D_{\mathrm{kl}}\left(P_{V_j}(\cdot \mid Y, V_{1:j-1}) \| P_{V_j}(\cdot \mid V_{1:j-1})\right) \mid V_{1:j-1}\right]\right]$$

$$\leq 4(e^{2\alpha} - 1)^2 \mathbb{E}_{V_{1:j-1}} \left[\mathbb{E}_Y \left[\left\|P_{X_j}(\cdot \mid V_{1:j-1}, Y) - P_{X_j}(\cdot \mid V_{1:j-1})\right\|_{\mathrm{TV}}^2 \mid V_{1:j-1}\right]\right].$$

Using Pinsker's inequality, we then find that

$$\mathbb{E}_{V_{1:j-1}} \left[\mathbb{E}_Y \left[\left\|P_{X_j}(\cdot \mid V_{1:j-1}, Y) - P_{X_j}(\cdot \mid V_{1:j-1})\right\|_{\mathrm{TV}}^2 \mid V_{1:j-1}\right]\right]$$

$$\leq \frac{1}{2}\mathbb{E}_{V_{1:j-1}} \left[\mathbb{E}_Y \left[D_{\mathrm{kl}}\left(P_{X_j}(\cdot \mid Y, V_{1:j-1}) \| P_{X_j}(\cdot \mid V_{1:j-1})\right) \mid V_{1:j-1}\right]\right] = \frac{1}{2}I(X_j; Y \mid V_{1:j-1}).$$

In particular, we have

$$I(V_j; Y \mid V_{1:j-1}) \leq 2\left(e^{2\alpha} - 1\right)^2 I(X_j; Y \mid V_{1:j-1}) \tag{25}$$

Lastly, we argue that $I(X_j; Y \mid V_{1:j-1}) \leq I(X_j; Y \mid X_{1:j-1})$. Indeed, we have by definition[2] that

$$I(X_j; Y \mid V_{1:j-1}) \overset{(i)}{=} H(X_j) - H(X_j \mid Y, V_{1:j-1})$$

$$\overset{(ii)}{\leq} H(X_j) - H(X_j \mid Y, V_{1:j-1}, X_{1:j-1})$$

$$\overset{(iii)}{=} H(X_j \mid X_{1:j-1}) - H(X_j \mid Y, X_{1:j-1}) = I(X_j; Y \mid X_{1:j-1}).$$

Here, equality $(i)$ follows since $X_j$ is independent of $V_{1:j-1}$, inequality $(ii)$ because conditioning reduces entropy, and equality $(iii)$ because $X_j$ is independent of $X_{1:j-1}$. Thus

$$I(V; Y) = \sum_{j=1}^d I(V_j; Y \mid V_{1:j-1}) \leq 2(e^{2\alpha} - 1)^2 \sum_{j=1}^d I(X_j; Y \mid X_{1:j-1}) = 2(e^{2\alpha} - 1)^2 I(X_{1:d}; Y),$$

which completes the proof.

## C   Proof of Theorem 2

In this section, we represent the $i$th sample by an $n_i$ sample matrix $X^{(i)} \in \mathbb{R}^{d \times n_i}$, where the $k$th column of $X^{(i)}$ is $X^{(i,k)}$ and $j$th row of $X^{(i)}$ is $X_j^{(i)}$. As usual, we assume the testing Markov chain $V \to X^{(i)} \to Y_i$, as in the setup for our proofs. We assume that $m \geq 4$, since otherwise the interactive lower bound (Proposition 1) provides a stronger result.

We have the following lemma, which is an analogue of Lemma 3.

**Lemma 4** *Let $V$ be sampled uniformly at random from $\{-1, 1\}^d$. For any $(i, j)$, assume that $X_j^{(i)}$ is independent of $\{X_{j'}^{(i)} : j' \neq j\} \cup \{V_{j'} : j' \neq j\}$ given $V_j$. Let $P_{X_j}$ be the probability measure of $X_j^{(i)}$ and assume in addition that*

$$\sup_{S \in \sigma(B_j)} \frac{P_{X_j}(S \mid V = \nu)}{P_{X_j}(S \mid V = \nu')} \leq \exp(\alpha).$$

*Define the random variable $E_j = 1$ if $X_j^{(i)} \in B_j$ and 0 otherwise. Then*

$$I(V; Y_i) \leq 2\left(e^{4\alpha} - 1\right)^2 I(X^{(i)}; Y_i) + \sum_{j=1}^{d} H(E_j) + \sum_{j=1}^{d} P(E_j = 0).$$

For the next lemma, we assume that as usual $\mathcal{V} = \{-1, 1\}^d$, and the parameter $\theta_\nu$ has coordinates given by $(\theta_\nu)_j = \nu_j \delta$. Moreover, we assume that each machine $i$ has $n_i$ independent samples from a $\mathsf{N}(\nu\delta, \sigma^2 I)$ distribution, so $\mathbb{E}_\nu[X] = \theta_\nu$. For conciseness we define the shorthand

$$b_i = \min\left\{128\frac{a^2}{\sigma^2}H(Y_i), d\right\}.$$

**Lemma 5** *Let $a > 0$ and $\delta > 0$ be chosen such that $\frac{\sqrt{n_i}a\delta}{\sigma^2} \leq \frac{1.2564}{4}$ for any $i \in \{1, \ldots, m\}$, and let $h(p) = -p\log(p) - (1-p)\log(1-p)$ be binary entropy. Then*

$$
\begin{aligned}
I(V; Y_i) &\leq \frac{n_i\delta^2}{\sigma^2}\min\left\{128\frac{a^2}{\sigma^2}H(Y_i), d\right\} + dh\left(2\exp\left(-\frac{(a - \sqrt{n_i}\delta)^2}{2\sigma^2}\right)\right) \\
&\quad + 2d\exp\left(-\frac{(a - \sqrt{n_i}\delta)^2}{2\sigma^2}\right).
\end{aligned}
\tag{26}
$$

With the bound (26) on the mutual information $I(Y_i; V)$, we may now divide our proof into two cases: when $d \geq 9$ and $d < 9$. Let us being with $d \geq 9$. Recalling our earlier minimax bound (16), we have—since $\theta(P_\nu) = \delta\nu$—that

$$\mathfrak{M}^{\mathrm{ind}}(\theta, \mathcal{P}, B_{1:m}) \geq \delta^2(\lfloor d/6 \rfloor + 1)\left(1 - \frac{I(Y_{1:m}; V) + \log 2}{d/6}\right).$$

If we can choose appropriate $\delta$ so that $I(Y_{1:m}; V) < 3/10$, then (since $d \geq 9$), we will obtain that the minimax error is lower bounded by $\delta^2(\lfloor d/6 \rfloor + 1)/2$, which will complete the proof.

Now, we consider each of the terms in the bound in Lemma 5 in turn, finding settings of $\delta$ and $a$ so that each is small. Specifically, recalling the assumption that $m \geq 2$, we will find settings of $\delta$ and $a$ so that the sum is bounded by $3/10$. We begin with the third term in the bound, where we note that if

$$\delta_3^2 \leq \frac{\sigma^2}{25 \cdot 16\log(m)\max_i n_i} \quad \text{and} \quad a = 5\sigma\sqrt{\log m}, \tag{27a}$$

then the condition $\frac{\sqrt{n_i}a\delta}{\sigma^2} \leq \frac{1.2564}{4}$ in Lemma 5 is satisfied. In addition, we have $(a - \sqrt{n_i}\delta_3)^2 \geq (5 - 1/20)^2\sigma^2\log m \geq 24\sigma^2\log m$, so

$$\sum_{i=1}^{m} 4\exp\left(-\frac{(a - \sqrt{n_i}\delta_3)^2}{2\sigma^2}\right) \leq 4m\exp\left(-12\log m\right) = \frac{4}{m^{11}} < 10^{-6}.$$

For the first term in the bound from Lemma 5, we note that with the identical choice of $a = 5\sigma\sqrt{n\log m}$, by taking

$$\delta_1^2 \leq \frac{d\sigma^2}{10\sum_{i=1}^{m} b_i n_i}, \tag{27b}$$

we have that $\sum_{i=1}^{m} 2b_i n_i \delta_1^2/(d\sigma^2) \leq 1/5$. Lastly, we have $h(q) \leq (6/5)\sqrt{q}$ for $q \geq 0$. As a consequence, we see that for $\delta_2^2$ chosen identically to the choice (27a) for $\delta_3$, we have

$$\sum_{i=1}^{m} 2h\left(2\exp\left(-\frac{(a - \sqrt{n_i}\delta_2)^2}{2\sigma^2}\right)\right) \leq \frac{12m}{5}\sqrt{2}\exp\left(-\frac{24}{4}\log m\right) \leq \frac{1}{300}.$$

In particular, combining bounds (27a) and (27b), we see that if we choose

$$\delta^2 = \min\left\{1, \frac{\sigma^2}{400\log(m)\max_i n_i}, \frac{d\sigma^2}{10\sum_{i=1}^{m} b_i n_i}\right\} \quad \text{and} \quad a = 5\sigma\sqrt{\log m},$$

then

$$\sum_{i=1}^{m} \frac{2b_i n_i \delta^2}{d\sigma^2} + 2h\left(2\exp\left(-\frac{(a-\sqrt{n_i}\delta)^2}{2\sigma^2}\right)\right) + 4\exp\left(-\frac{(a-\sqrt{n_i}\delta)^2}{2\sigma^2}\right) < \frac{3}{10}.$$

This completes the proof for the case that $d \geq 9$, since

$$b_i \leq \min\left\{128\frac{a^2}{\sigma^2}H(Y_i), d\right\} = \min\left\{25 \cdot 128 H(Y_i)\log m, d\right\} \leq \min\left\{25 \cdot 128 B_i \log m, d\right\}$$

by Shannon's source coding theorem.

When $d < 9$, an appeal to Le Cam's method (20), as in the proof of Proposition 2, and an identical series of steps to bound the mutual information using inequality (26) (i.e., again applying inequalities (27a)–(27b)) completes the proof.

### C.1  Proof of Lemma 4

The proof is substantially similar to that of Lemma 3, but we exhibit some care since we must condition on the event that $X_j^{(i)} \in B_j$. For notational simplicity, we again suppress all dependence of $X$ and $Y$ on the machine index $i$.

We begin by noting that given $E_j$, the variable $V_j$ is independent of $V_{1:j-1}$, $X_{1:j-1}$, $V_{j+1:d}$, and $X_{j+1:d}$. Moreover, by the assumption in the lemma we have for any $S \in \sigma(B_j)$ that

$$\frac{P_{X_j}(S \mid V = \nu, E_j = 1)}{P_{X_j}(S \mid V = \nu', E_j = 1)} = \frac{P_{X_j}(S \mid V = \nu)}{P_{X_j}(X_j \in B_j \mid V = \nu)} \frac{P_{X_j}(X_j \in B_j \mid V = \nu')}{P_{X_j}(X_j \in S \mid V = \nu')} \leq \exp(2\alpha),$$

so we have the analogue of the bound (24) that

$$P(V_j = \nu_j \mid V_{1:j-1}, Y, E_j = 1) - P(V_j = \nu_j \mid V_{1:j-1}, E_j = 1)$$
$$\leq 2\left(e^{4\alpha} - 1\right)\left\|P_{X_j}(\cdot \mid V_{1:j-1}, Y, E_j = 1) - P_{X_j}(\cdot \mid V_{1:j-1}, E_j = 1)\right\|_{\mathrm{TV}} \cdot \dots \qquad (28)$$
$$\min\left\{P(V_j = \nu_j \mid V_{1:j-1}, Y, E_j = 1), P(V_j = \nu_j \mid V_{1:j-1}, E_j = 1)\right\}.$$

Thus, proceeding as in the proof of Lemma 3 (specifically the argument preceding inequality (25)), the expression (28) implies

$$I(V_j; Y \mid V_{1:j-1}, E_j = 1) \leq 2\left(e^{4\alpha} - 1\right)^2 I(X_j; Y \mid V_{1:j-1}, E_j = 1). \qquad (29)$$

The bound (29) as stated conditions on $E_j$, which makes it somewhat unwieldy. We turn to removing this conditioning. By the definition of (conditional) mutual information, we have

$$P(E_j = 1)I(V_j; Y \mid V_{1:j-1}, E_j = 1)$$
$$= I(V_j; Y \mid V_{1:j-1}, E_j) - I(V_j; Y \mid V_{1:j-1}, E_j = 0)P(E_j = 0)$$
$$= I(V_j; E_j, Y \mid V_{1:j-1}) - I(V_j; E_j \mid V_{1:j-1}) - I(V_j; Y \mid V_{1:j-1}, E_j = 0)P(E_j = 0)$$

Since conditioning reduces entropy,

$$I(V_j; E_j, Y \mid V_{1:j-1}) = H(V_j \mid V_{1:j-1}) - H(V_j \mid E_j, Y, V_{1:j-1})$$
$$\geq H(V_j \mid V_{1:j-1}) - H(V_j \mid Y, V_{1:j-1}) = I(V_j; Y \mid V_{1:j-1}),$$

and noting that $I(V_j; Y \mid V_{1:j-1}, E_j = 0) \leq H(V_j) \leq 1$ and $I(V_j; E_j \mid V_{1:j-1}) \leq H(E_j)$ gives

$$P(E_j = 1)I(V_j; Y \mid V_{1:j-1}, E_j = 1) \geq I(V_j; Y \mid V_{1:j-1}) - H(E_j) - P(E_j = 0). \qquad (30)$$

We now combine inequalities (30) and (29) to complete the proof of the lemma. By the definition of conditional mutual information,

$$I(X_j; Y \mid V_{1:j-1}, E_j = 1) \leq \frac{I(X_j; Y \mid V_{1:j-1}, E_j)}{P(E_j = 1)} \leq \frac{I(X_j; Y \mid V_{1:j-1})}{P(E_j = 1)}.$$

Combining this with inequalities (30) and (29) yields

$$I(V_j; Y \mid V_{1:j-1}) \leq H(E_j) + P(E_j = 0) + 2\left(e^{4\alpha} - 1\right)^2 I(X_j; Y \mid V_{1:j-1}).$$

Up to the additive terms, this is equivalent to the earlier bound (25) in the proof of Lemma 3; proceeding *mutatis mudandis* we complete the proof.

## C.2 Proof of Lemma 5

Inequality (26) is the consequence of two intermediate upper bounds, which we prove separately:

$$I(V; Y_i) \le \frac{d n_i \delta^2}{\sigma^2}, \tag{31}$$

$$I(V; Y_i) \le 128 \frac{\delta^2 a^2}{\sigma^4} n_i H(Y_i)$$
$$+ dh \left( 2 \exp\left( -\frac{(a - \sqrt{n_i}\delta)^2}{2\sigma^2} \right) \right) + 2d \exp\left( -\frac{(a - \sqrt{n_i}\delta)^2}{2\sigma^2} \right). \tag{32}$$

To prove inequality (31), we note that $V \to X^{(i)} \to Y_i$ forms a Markov chain. Thus, the data-processing inequality [6] implies that

$$I(V; Y_i) \le I(V; X^{(i)}) \le \sum_{j=1}^{n_i} I(V; X^{(i,j)}) = n_i I(V; X^{(i,1)})$$

where the last inequality comes from the independence of the $X^{(i,j)}$. Let $P_\nu$ denote the conditional distribution of $X^{(i,j)}$ given $V = \nu$. Then the convexity of the KL-divergence implies

$$I(V; X^{(i,j)}) \le \frac{1}{|\mathcal{V}|^2} \sum_{\nu, \nu' \in \mathcal{V}} D_{\mathrm{kl}}\left( P_\nu \| P_{\nu'} \right) = \frac{\delta^2}{2\sigma^2} \frac{1}{|\mathcal{V}|^2} \sum_{\nu, \nu' \in \mathcal{V}} \| \nu - \nu' \|_2^2 = \frac{d\delta^2}{\sigma^2}.$$

This establishes inequality (31).

To prove inequality (32), we apply Lemma 4. First, we note that by taking a ratio of the densities of two normals with $n_i$ independent samples, one with mean $\delta$ and the other with mean $-\delta$, both with variance $\sigma^2$, we have

$$\frac{\exp(-\frac{1}{2\sigma^2} \sum_{l=1}^{n_i} (x_l - \delta)^2)}{\exp(-\frac{1}{2\sigma^2} \sum_{l=1}^{n_i} (x_l + \delta)^2)} = \exp\left( \frac{2\delta}{2\sigma^2} \sum_{l=1}^{n_i} x_l \right) \le \exp\left( \frac{\sqrt{n_i}\delta a}{\sigma^2} \right)$$

whenever $|\sum_l x_l| \le \sqrt{n_i} a$. As a consequence, we see that by taking the sets

$$B_j = \left\{ x \in \mathbb{R}^{n_i} : \left| \sum_{l=1}^{n_i} x_l \right| \le \sqrt{n_i} a \right\},$$

we satisfy the conditions of Lemma 4 with $\alpha = \sqrt{n_i}\delta a / \sigma^2$. In addition, when $\alpha \le 1.2564$, we have $\exp(\alpha) - 1 \le 2\alpha$, so under the conditions of the lemma, $\exp(4\alpha) - 1 = \exp(4\sqrt{n_i}\delta a / \sigma^2) - 1 \le 8\sqrt{n_i}\delta a / \sigma^2$. Recalling the definition of the event $E_j = \{X_j^{(i)} \in B_j\}$ from Lemma 4, we obtain

$$I(V; Y_i) \le 128 \frac{\delta^2 a^2}{\sigma^4} n_i I(X^{(i)}; Y_i) + \sum_{j=1}^{d} H(E_j) + \sum_{j=1}^{d} P(E_j = 0).$$

Comparing this inequality with inequality (32), we see that we must bound the probability of the event $E_j = 0$.

Bounding $P(E_j = 0)$ is not challenging, however. From standard Gaussian tail bounds, we have for $Z_i$ distributed i.i.d. according to $\mathsf{N}(\delta, \sigma^2)$ that

$$P(E_j = 0) = P\left( \left| \sum_{l=1}^{n_i} Z_l \right| \ge \sqrt{n_i} a \right)$$
$$= P\left( \sum_{l=1}^{n_i} (Z_l - \delta) \ge \sqrt{n_i} a - n\delta \right) + P\left( \sum_{l=1}^{n_i} (Z_l - \delta) \le \sqrt{n_i} a - n\delta \right)$$
$$\le 2 \exp\left( -\frac{(a - \sqrt{n_i}\delta)^2}{2\sigma^2} \right).$$

# D   Proof of Proposition 1

We prove the lower bound via a standard information-theoretic argument. Fix $\delta > 0$, and let $\mathcal{V} = [2^{\log M_\Theta(2\delta)}]$ index a maximal $2\delta$-packing of $\Theta$, which we identify by $\{\theta_\nu\}_{\nu \in \mathcal{V}} \subset \Theta$. Fix an (arbitrary) protocol $\Pi$ for communication.

Following the standard reduction from (worst-case) estimation to testing [20, 22, 19], let $V$ be sampled uniformly from $\mathcal{V}$. For messages $Y = (Y_1, \ldots, Y_T)$ sent by the protocol $\Pi$, let $\widehat{\theta}(Y)$ denote the estimator of $\theta$ based on $Y$ and define $\widehat{V} = \operatorname{argmin}_{\nu \in \mathcal{V}} \|\widehat{\theta}(Y) - \theta_\nu\|_2$. Then $\|\widehat{\theta}(Y) - \theta_\nu\|_2 \geq \delta$ if $\widehat{V} \neq V$, and we have

$$\max_{\nu \in \mathcal{V}} \mathbb{E}\left[\|\widehat{\theta}(Y) - \theta_\nu\|_2^2\right] \geq \sum_{\nu \in \mathcal{V}} \mathbb{P}(V = \nu)\mathbb{E}\left[\|\widehat{\theta}(Y) - \theta_V\|_2^2 \mid V = \nu\right]$$

$$\geq \sum_{\nu \in \mathcal{V}} \delta^2 \mathbb{P}(V = \nu)\mathbb{P}(\widehat{V} \neq V \mid V = \nu) = \delta^2 \mathbb{P}(\widehat{V} \neq V). \quad (33)$$

By Fano's inequality [6], the testing error (33) is lower bounded by

$$\mathbb{P}(\widehat{V} \neq V) \geq 1 - \frac{I(V;Y) + 1}{\log M_\Theta(2\delta)} \geq 1 - \frac{H(Y) + 1}{\log M_\Theta(2\delta)},$$

since $H(Y) \geq I(V;Y)$. Shannon's source coding theorem [6, Chapter 5] guarantees the lower bound $B \geq H(Y)$. Since the protocol $\Pi$ was arbitrary, we have as an immediate consequence of inequality (33) that

$$\mathfrak{M}^{\text{inter}}(\theta, \mathcal{P}, B) \geq \delta^2 \left(1 - \frac{B + 1}{\log M_\Theta(2\delta)}\right) \quad \text{for any } \delta \geq 0. \quad (34)$$

Using inequality (34), the remainder of the proof is straightforward. Indeed, we have

$$1 - \frac{B + 1}{\log M_\Theta(2\delta)} \geq \frac{1}{2} \text{ iff } \frac{\log M_\Theta(2\delta)}{B + 1} \geq 2 \text{ iff } 2\delta \geq \log M_\Theta^{-1}(2B + 2).$$

Setting $\delta = \frac{1}{2}\log M_\Theta^{-1}(2B + 2)$ thus gives the result of the theorem.

# E   Proof of Theorem 1

We follow a standard hypothesis testing setup (recall Section 5.1) to choose a variable $V \in \{-1, 1\}$ uniformly at random and then sample $X^{(i)}$ w.r.t. $\mathsf{N}(\delta V, \sigma^2)$ independently on each of the $m$ machines. However, in this situation, while the local samples are independent, the messages are not: the sequence of random variables $Y = (Y_1, \ldots, Y_T)$ is generated such that the distribution of $Y_t$ is a measurable function of $(X^{(i_t)}, Y_{1:t-1})$ where $i_t \in \{1, \ldots, m\}$ is the index the existing sample upon which $Y_t$ is based. We assume without loss of generality that the sequence $\{i_1, i_2, \ldots, \}$ is fixed in advance—if the choice of index $i_t$ is based on $Y_{1:t-1}$ and $X$, then we simply say there exists a default value (say $Y_t = \perp$) that indicates "nothing."

**Lemma 6** *Assume that $|\mathcal{V}| = 2$. Also assume that there is a set $B$ such that for any $\nu, \nu' \in \mathcal{V}$ we have*

$$\sup\left\{\frac{P_{X^{(i)}}(S \mid \nu)}{P_{X^{(i)}}(S \mid \nu')}\middle| S \in \sigma(B), \nu, \nu' \in \mathcal{V}\right\} \leq e^\alpha. \quad (35)$$

*Let the random variable $\mathcal{E} = 1$ if $X^{(i)} \in B$ for all $i$ and $\mathcal{E} = 0$ otherwise. Then*

$$I(V;Y) \leq 2\left(e^{4\alpha} - 1\right)^2 I(X;Y) + H(\mathcal{E}) + P(\mathcal{E} = 0).$$

Consider the following scheme. Given $\nu \in \{-1, 1\}$, we assume that each machine $i$ receives $n$ sample $X^{(i,k)}$ ($k = 1, \ldots, n$) independently sampled according to

$$X \sim \mathsf{N}(\delta\nu, \sigma^2)$$

Following the low dimension case of Proposition 2, inequality (23) implies that

$$\text{if } I(V;Y) \leq \frac{3}{10} \text{ then } \sup_{\theta \in \Theta} \mathbb{E}[(\widehat{\theta} - \theta)^2] > \frac{\delta^2}{10}. \tag{36}$$

We focus on showing the conditions for the implication (36) hold. By defining $B = \{x \in \mathbb{R}^n : |\sum_{i=1}^n x_i \leq \sqrt{n}a\}$ and the condition of Lemma 6 is satisfied with $\alpha = \sqrt{n}\delta a / \sigma^2$. If we assume that $\alpha \leq 1.2564$ (which is satisfied by the assignment described below), then $\exp(\alpha) - 1 \leq 2\alpha$ and hence $\exp(4\alpha) - 1 = \exp(4\sqrt{n}\delta a / \sigma^2) - 1 \leq 8\sqrt{n}\delta a / \sigma^2$. We obtain

$$I(V;Y) \leq 128 \frac{\delta^2 n a^2}{\sigma^4} H(Y) + H(\mathcal{E}) + P(\mathcal{E} = 0).$$

Let $E_i$ be the random variable such that $E_i = 1$ if $X^{(i)} \in B$ and $E_i = 0$ otherwise. Since $\mathcal{E} = \prod_{i=1}^m E_i$, we have $P(\mathcal{E} = 0) \leq \sum_{i=1}^m P(\mathbb{E}_i = 0)$. We apply the last inequality in the proof of Lemma 5 to upper bounds $P(E_i = 0)$, which yields that

$$P(\mathcal{E} = 0) \leq \sum_{i=1}^m P(E_i = 0) \leq 2m \exp\left(-\frac{(a - \sqrt{n}\delta)^2}{2\sigma^2}\right).$$

Consequently,

$$I(V;Y) \leq 128 \frac{\delta^2 n a^2}{\sigma^4} H(Y) + mh\left(2\exp\left(-\frac{(a - \sqrt{n}\delta)^2}{2\sigma^2}\right)\right) + 2m \exp\left(-\frac{(a - \sqrt{n}\delta)^2}{2\sigma^2}\right), \tag{37}$$

where $h(p) = -p\log(p) - (1 - p)\log(1 - p)$ is the binary entropy function. We also used the convexity of $h$ in $[0, 1/2]$, so that $h(p) \leq mh(p/m)$ for $0 \leq p \leq 1/2$.

Given upper bound (37), we follow the proof of Theorem 2 to see that by choosing

$$\delta^2 = \min\left\{1, \frac{\sigma^2}{400 \log(m)n}, \frac{\sigma^2}{10 \cdot 128 \cdot 36 \log(m)nH(Y)}\right\} \quad \text{and} \quad a = 5\sigma\sqrt{\log m},$$

we obtain $I(V;Y) \leq \frac{3}{10}$. Thus, there is a universal constant $c$ such that

$$\max_{\nu \in \mathcal{V}} \mathbb{E}[(\widehat{\theta} - \theta)^2] > c \min\left\{1, \frac{\sigma^2}{\log(m)n}, \frac{\sigma^2}{\log(m)nH(Y)}\right\}.$$

Applying the source coding theorem to bound $H(Y) \leq B$ completes the proof.

## E.1 Proof of Lemma 6

**Lemma 7** *Consider the hypothesis testing problem described in the second paragraph of Appendix E, but assume that $X^{(i)}$ is sampled from another probability measure $Q(\cdot \mid \nu)$ satisfying*

$$\sup\left\{\frac{Q(S \mid \nu)}{Q(S \mid \nu')} \mid S \in \sigma(\mathcal{X}), \nu, \nu' \in \mathcal{V}\right\} \leq e^\alpha. \tag{38}$$

*Then we have*

$$I(V;Y) \leq 2\left(e^{2\alpha} - 1\right)^2 I(X;Y).$$

With Lemma 7 established, the proof of Lemma 6 follows, *mutatis mutandis*, as in the proof of Lemma 4 from Lemma 3. Thus, it only remains to prove Lemma 7.

**Proof of Lemma 7** By the chain-rule for mutual information, we have that

$$I(V;Y) = \sum_{t=1}^T I(V;Y_t \mid Y_{1:t-1}).$$

Let $P_{Y_t}(\cdot \mid Y_{1:t-1})$ denote the (marginal) distribution of $Y_t$ given $Y_{1:t-1}$ and define $P_V(\cdot \mid Y_{1:t})$ to be the distribution of $V$ conditional on $Y_{1:t}$. Then we have by marginalization that

$$P_V(\cdot \mid Y_{1:t-1}) = \int P_V(\cdot \mid Y_{1:t-1}, y_t)dP_{Y_t}(y_t \mid Y_{1:t-1})$$

and thus

$$I(V; Y_t \mid Y_{1:t-1}) = \mathbb{E}_{Y_{1:t-1}}\left[\mathbb{E}_{Y_t}\left[D_{\mathrm{kl}}\left(P_V(\cdot \mid Y_{1:t}) \| P_V(\cdot \mid Y_{1:t-1})\right) \mid Y_{1:t-1}\right]\right]. \tag{39}$$

We now bound the above KL divergence using the assumptions in the lemma.

By the nonnegativity of the KL divergence, we have

$$\begin{aligned}
D_{\mathrm{kl}}&\left(P_V(\cdot \mid Y_{1:t}) \| P_V(\cdot \mid Y_{1:t-1})\right) \\
&\leq D_{\mathrm{kl}}\left(P_V(\cdot \mid Y_{1:t}) \| P_V(\cdot \mid Y_{1:t-1})\right) + D_{\mathrm{kl}}\left(P_V(\cdot \mid Y_{1:t-1}) \| P_V(\cdot \mid Y_{1:t})\right) \\
&= \sum_{\nu \in \mathcal{V}} \left(p_V(\nu \mid Y_{1:t-1}) - p_V(\nu \mid Y_{1:t})\right) \log \frac{p_V(\nu \mid Y_{1:t-1})}{p_V(\nu \mid Y_{1:t})}
\end{aligned}$$

where $p_V$ denotes the p.m.f. of $V$. We claim that Lemma 8 implies that

$$\begin{aligned}
|p_V&(\nu \mid Y_{1:t-1}) - p_V(\nu \mid Y_{1:t})| \\
&\leq 2\left(e^{2n\alpha} - 1\right) \min\left\{p_V(\nu \mid Y_{1:t-1}), p_V(\nu \mid Y_{1:t})\right\} \|P_{X^{(i_t)}}(\cdot \mid Y_{1:t}) - P_{X^{(i_t)}}(\cdot \mid Y_{1:t-1})\|_{\mathrm{TV}}.
\end{aligned} \tag{40}$$

Deferring the proof of inequality (40) to the end of this section, we give the remainder of the proof. First, by a first-order convexity argument, we have that for any $a, b > 0$

$$\log \frac{a}{b} \leq \frac{|a-b|}{\min\{a,b\}}.$$

As a consequence, we find

$$\begin{aligned}
\left(p_V(\nu \mid Y_{1:t-1}) - p_V(\nu \mid Y_{1:t})\right) &\log \frac{p_V(\nu \mid Y_{1:t-1})}{p_V(\nu \mid Y_{1:t})} \leq \frac{\left(p_V(\nu \mid Y_{1:t-1}) - p_V(\nu \mid Y_{1:t})\right)^2}{\min\{p_V(\nu \mid Y_{1:t-1}), p_V(\nu \mid Y_{1:t})\}} \\
&\leq 4\left(e^{2n\alpha} - 1\right)^2 \min\left\{p_V(\nu \mid Y_{1:t-1}), p_V(\nu \mid Y_{1:t})\right\} \|P_{X^{(i_t)}}(\cdot \mid Y_{1:t}) - P_{X^{(i_t)}}(\cdot \mid Y_{1:t-1})\|_{\mathrm{TV}}^2
\end{aligned}$$

by using inequality (40). Using the fact that $p_V$ is a p.m.f., we thus have

$$\begin{aligned}
D_{\mathrm{kl}}&\left(P_V(\cdot \mid Y_{1:t}) \| P_V(\cdot \mid Y_{1:t-1})\right) + D_{\mathrm{kl}}\left(P_V(\cdot \mid Y_{1:t-1}) \| P_V(\cdot \mid Y_{1:t})\right) \\
&\leq 4\left(e^{2n\alpha} - 1\right)^2 \|P_{X^{(i_t)}}(\cdot \mid Y_{1:t}) - P_{X^{(i_t)}}(\cdot \mid Y_{1:t-1})\|_{\mathrm{TV}}^2 \sum_{\nu \in \mathcal{V}} \min\left\{p_V(\nu \mid Y_{1:t-1}), p_V(\nu \mid Y_{1:t})\right\} \\
&\leq 4\left(e^{2n\alpha} - 1\right)^2 \|P_{X^{(i_t)}}(\cdot \mid Y_{1:t}) - P_{X^{(i_t)}}(\cdot \mid Y_{1:t-1})\|_{\mathrm{TV}}^2.
\end{aligned}$$

Using Pinsker's inequality, we then find that

$$\begin{aligned}
\mathbb{E}_{Y_{1:t-1}}&\left[\mathbb{E}_{Y_t}\left[\|P_{X^{(i_t)}}(\cdot \mid Y_{1:t}) - P_{X^{(i_t)}}(\cdot \mid Y_{1:t-1})\|_{\mathrm{TV}}^2 \mid Y_{1:t-1}\right]\right] \\
&\leq \frac{1}{2}\mathbb{E}_{Y_{1:t-1}}\left[\mathbb{E}_{Y_t}\left[D_{\mathrm{kl}}\left(P_{X^{(i_t)}}(\cdot \mid Y_{1:t}) \| P_{X^{(i_t)}}(\cdot \mid Y_{1:t-1})\right) \mid Y_{1:t-1}\right]\right] = \frac{1}{2}I(X^{(i_t)}; Y_t \mid Y_{1:t-1}).
\end{aligned}$$

Since conditioning reduces entropy and $Y$ is discrete, we have

$$\begin{aligned}
I(X^{(i_t)}; Y_t \mid Y_{1:t-1}) &= H(Y_t \mid Y_{1:t-1}) - H(Y_t \mid X^{(i_t)}, Y_{1:t-1}) \\
&\leq H(Y_t \mid Y_{1:t-1}) - H(Y_t \mid X, Y_{1:t-1}) = I(X; Y_t \mid Y_{1:t-1}).
\end{aligned}$$

This completes the proof of the lemma, since $\sum_{t=1}^{T} I(X; Y_t \mid Y_{1:t-1}) = I(X; Y)$ by the chain rule for information.

**Proof of inequality** (40)   To establish the inequality, we give a one-to-one correspondence between the variables in Lemma 8 and the variables in Lemma 7. We make the following identifications:

$$V \leftrightarrow A \qquad X^{(i_t)} \leftrightarrow B \qquad Y_{1:t-1} \leftrightarrow C \qquad Y_t \leftrightarrow D.$$

For Lemma 8 to hold, we must verify conditions (43), (44), and (45). For condition (43) to hold, $Y_t$ must be independent of $V$ given $\{Y_{1:t-1}, X^{(i_t)}\}$. Since the distribution of $P_{Y_t}(\cdot \mid Y_{1:t-1}, X^{(i_t)})$ is measurable-$\{Y_{1:t-1}, X^{(i_t)}\}$, Condition (45) is satisfied by the assumption in the lemma.

Finally, for condition (44) to hold, we must be able to factor the conditional probability of $Y_{1:t-1}$ given $\{V, X^{(i_t)}\}$ as

$$P(Y_{1:t-1} = y_{1:t-1} \mid V, X^{(i_t)}) = \Psi_1(V, y_{1:t-1})\Psi_2(X^{(i_t)}, y_{1:t-1}). \tag{41}$$

To prove this decomposition, notice that

$$P(Y_{1:t-1} = y_{1:t-1} \mid V, X^{(i_t)}) = \prod_{k=1}^{t-1} P(Y_k = y_k \mid Y_{1:k-1}, V, X^{(i_t)}).$$

For any $k \in \{1, \ldots, t-1\}$, if $i_k = i_t$—that is, the message $Y_k$ is generated based on sample $X^{(i_t)} = X^{(i_k)}$—then $Y_k$ is independent of $V$ given $\{X^{(i_t)}, Y_{1:k-1}\}$. Thus, $P_{Y_k}(\cdot \mid Y_{1:k-1}, V, X^{(i_t)})$ is measurable-$\{X^{(i_t)}, Y_{1:k-1}\}$. If the $k$th index $i_k \neq i_t$, then $Y_k$ is independent of $X^{(i_t)}$ given $\{Y_{1:k-1}, V\}$ by construction, which means $P_{Y_k}(\cdot \mid Y_{1:k-1}, V, X^{(i_t)}) = P_{Y_k}(\cdot \mid Y_{1:k-1}, V)$. The decomposition (41) thus holds, and we have verified that each of the conditions of Lemma 8 holds. We thus establish inequality (40).

# F    Proof of Corollary 1

We prove Corollary 1 in two parts: the upper bound (for part (a)) and lower bound (for part (b)). We prove the upper bound by exhibiting an interactive protocol $\Pi^*$ and prove the lower bound by applying Proposition 1.

**Upper bound on the minimax risk**    We consider the following communication protocol $\Pi^* \in \mathcal{A}_{\text{inter}}(B, \mathcal{P})$:

1. Machine $i \in \{1, \ldots, m\}$ computes its local minimum $a^{(i)} = \min\{X^{(i,k)} : k = 1, \ldots, n\}$.

2. Machine 1 broadcasts $a^{(1)}$ using $2\log(mn)$ bits. Upon receiving the broadcast, all machines initialize global minimum variables $s \leftarrow a^{(1)}$.

3. In the order $i = 2, 3, \ldots, m$, machine $i$ performs the following operations:

    (i) Check if $a^{(i)} < s$. If so, machine $i$ performs the update $s \leftarrow a^{(i)}$ and broadcasts $s$, otherwise it does nothing.

    (ii) All other machines update their local $s$ after receiving machine $i$'s update. All real numbers in the message are rounded down to $2\log(mn)$-bit discrete values.

4. One machine outputs $\widehat{\theta} = s + 1$.

According to the protocol described above, $\Pi^*$ computes a global minima

$$s = \min\left\{X^{(i,k)} : i = 1, \ldots, m;\ k = 1, \ldots, n\right\}$$

to accuracy of $\mathcal{O}(1/(mn)^2)$ since because real numbers are encoded with $2\log(mn)$ bits. Then classical convergence analysis [14] yields estimator $\widehat{\theta} = s + 1$ achieves minimax optimal convergence rate $\mathbb{E}[\|\widehat{\theta} - \theta\|_2^2] \lesssim 1/(mn)^2$.

To analyze the communication complexity of the protocol $\Pi^*$, we study Steps 2–3. In Step 2, machine 1 sends $2\log(mn)$ bits as message $Y_1$. In Step 3, machine $i$ sends $2\log(mn)$ bits only if $a^{(i)} < \min\{a^{(1)}, \cdots, a^{(i-1)}\}$. By inspection, this event happens with probability bounded by $1/i$, so we find that the expected length of message $Y_i$ is

$$\mathbb{E}[L_i] \leq \frac{2\log(mn)}{i}.$$

Putting all pieces together, we obtain that

$$\mathbb{E}[L] = \sum_{i=1}^{m} \mathbb{E}[L_i] \leq 2\log(mn) + \sum_{i=2}^{m} \frac{2\log(dmn)}{i} \leq 2\log(mn) + 2\ln(m)\log(mn).$$

Figure 1: Graphical model for Lemma 8

**Lower bound on the minimax risk**  To prove the lower bound, we simply evaluate packing entropies by using a volume argument [3]. Since $\Theta = [-1, 1]$, the size of a maximal $2\delta$-packing can be lower bounded by

$$2^{\log M_\Theta(2\delta)} \geq \frac{\text{Volume}(\Theta)}{\text{Volume}(\{x \in \mathbb{R} : \|x\|_2 \leq 2\delta\})} \geq \frac{1}{2\delta}. \tag{42}$$

Taking logarithms and inverting $B = \log M_\Theta(\delta) = \log M_\Theta(1/(mn))$ yields the lower bound.

## G  Total variation contraction

In this section, we prove a technical lemma that is essential to the proof of our results.

Consider four random variables $A, B, C, D$, of which we assume that $A$, $C$, and $D$ have discrete distributions. We denote the conditional distribution of $A$ given $B$ by $P_{A|B}$ and their full joint distribution by $P_{A,B,C,D}$. We assume that the random variables have conditional indpendence structure specified by the graphical model in Figure 1, that is, that we can write the joint distribution as the product

$$P_{A,B,C,D} = P_A P_{B|A} P_{C|A,B} P_{D|B,C}. \tag{43}$$

We denote the domain of a random variable by the identical calligraphic letter, so $A \in \mathcal{A}$, $B \in \mathcal{B}$, and so on. We write $\sigma(\mathcal{A})$ for the sigma-field on $\mathcal{A}$ with respect to which our measures are defined. Sometimes we write $P_A(\cdot \mid B)$ for the conditional distribution of $A$ given $B$. In addition to the conditional independence assumption (43), we assume that the conditional distribution of $C$ given $A, B$ factorizes in the following specific form. There exist functions $\Psi_1 : \mathcal{A} \times \sigma(\mathcal{C}) \to \mathbb{R}_+$ and $\Psi_2 : \mathcal{B} \times \sigma(\mathcal{C}) \to \mathbb{R}_+$ such that for any (measureable) set $S$ in the range $\mathcal{C}$ of $C$, we have

$$P_C(S \mid A, B) = \Psi_1(A, S)\Psi_2(B, S). \tag{44}$$

Since $C$ is assumed discrete, we abuse notation and write $P(C = c \mid A, B) = \Psi_1(A, c)\Psi_2(B, c)$. Lastly, we assume that for any $a, a' \in \mathcal{A}$, we have the following likelihood ratio bound:

$$\sup_{S \in \sigma(\mathcal{B})} \frac{P_B(S \mid A = a)}{P_B(S \mid A = a')} \leq \exp(\alpha). \tag{45}$$

**Lemma 8** *Under assumptions* (43), (44), *and* (45), *the following inequality holds:*

$$|P(A = a \mid C, D) - P(A = a \mid C)|$$
$$\leq 2\left(e^{2\alpha} - 1\right) \min\left\{P(A = a \mid C), P(A = a \mid C, D)\right\} \|P_B(\cdot \mid C, D) - P_B(\cdot \mid C)\|_{\text{TV}}.$$

**Proof:**  By assumption, $A$ is independent of $D$ given $\{B, C\}$. Thus we may write

$$P(A = a \mid C, D) - P(A = a \mid C) = \int P(A = a \mid B = b, C)\left(dP_B(b \mid C, D) - dP_B(b \mid C)\right)$$

Combining this equation with the inequality

$$\int P(A = a \mid C)\left(dP_B(b \mid C, D) - dP_B(b \mid C)\right) = 0$$

we find that

$$P(A = a \mid C, D) - P(A = a \mid C)$$
$$= \int \left( P(A = a \mid B = b, C) - P(A = a \mid C) \right) \left( dP_B(b \mid C, D) - dP_B(b \mid C) \right).$$

Using the fact that $\left| \int f(b) d\mu(b) \right| \leq \sup_b \{|f(b)|\} \int |d\mu(b)|$ for any signed measure $\mu$ on $\mathcal{B}$, we conclude from the previous equality that for *any* version $P_A(\cdot \mid B, C)$ of the conditional probability of $A$ given $\{B, C\}$ that since $\int |d\mu| = \|\mu\|_{\mathrm{TV}}$,

$$|P(A = a \mid C, D) - P(A = a \mid C)|$$
$$\leq 2 \sup_{b \in \mathcal{B}} \{|P(A = a \mid B = b, C) - P(A = a \mid C)|\} \, \|P_B(\cdot \mid C, D) - P_B(\cdot \mid C)\|_{\mathrm{TV}}.$$

Thus, to prove the lemma, it is sufficient to show (for some version of the conditional distribution[3] $P_A(\cdot \mid B, C)$) that for any $b \in \mathcal{B}$

$$|P(A = a \mid B = b, C) - P(A = a \mid C)| \leq (e^{2\alpha} - 1) \min\{P(A = a \mid C), P(A = a \mid C, D)\}. \tag{46}$$

To prove this upper bound, we consider the joint distribution (43) and likelihood ratio bound (46). The distributions $\{P_B(\cdot \mid A = a)\}_{a \in \mathcal{A}}$ are all absolutely continuous with respect to one another by assumption (46), so it is no loss of generality to assume that there exists a density $p_B(\cdot \mid A = a)$ for which $P(B \in S \mid A = a) = \int p_B(b \mid A = a) d\mu(b)$, for some fixed measure $\mu$, and for which the ratio $p_B(b \mid A = a)/p_B(b \mid A = a') \in [e^{-\alpha}, e^{\alpha}]$ for all $b$. By elementary conditioning we have for any $S_b \in \sigma(\mathcal{B})$ and $c \in \mathcal{C}$

$$P(A = a \mid B \in S_b, C = c)$$
$$= \frac{P(A = a, B \in S_b, C = c)}{P(B \in S_b, C = c)}$$
$$= \frac{P(B \in S_b, C = c \mid A = a) P(A = a)}{\sum_{a' \in \mathcal{A}} P(A = a') P(B \in S_b, C = c \mid A = a)}$$
$$= \frac{P(A = a) \int_{S_b} P(C = c \mid B = b, A = a) p_B(b \mid A = a) d\mu(b)}{\sum_{a' \in \mathcal{A}} P(A = a') \int_{S_b} P(C = c \mid B = b, A = a') p_B(b \mid A = a') d\mu(b)},$$

where for the last equality we used the conditional independence assumptions (43). But now we recall the decomposition formula (44), and we can express the likelihood functions by

$$P(A = a \mid B \in S_b, C = c) = \frac{P(A = a) \int_{S_b} \Psi_1(a, c) \Psi_2(b, c) p_B(b \mid A = a) d\mu(b)}{\sum_{a'} P(A = a') \int_{S_b} \Psi_1(a', c) \Psi_2(b, c) p_B(b \mid A = a') d\mu(b)}.$$

As a consequence, there is a version of the conditional distribution of $A$ given $B$ and $C$ such that

$$P(A = a \mid B = b, C = c) = \frac{P(A = a) \Psi_1(a, c) p_B(b \mid A = a)}{\sum_{a'} P(A = a') \Psi_1(a', c) p_B(b \mid A = a')}. \tag{47}$$

Define the shorthand

$$\beta = \frac{P(A = a) \Psi_1(a, c)}{\sum_{a' \in \mathcal{A}} P(A = a') \Psi_1(a', c)}.$$

We claim that

$$e^{-\alpha} \beta \leq P(A = a \mid B = b, C = c) \leq e^{\alpha} \beta. \tag{48}$$

Assuming the correctness of bound (48), we establish inequality (46). Indeed, since $P(A = a \mid C = c)$ is a weighted average of $P(A = a \mid B = b, C = c)$, we also have the same upper and lower bound for $P(A = a \mid C)$, that is

$$e^{-\alpha} \beta \leq P(A = a \mid C) \leq e^{\alpha} \beta,$$

while the conditional independence assumption that $A$ is independent of $D$ given $B, C$ (recall Figure 1 and the product (43)) implies

$$
\begin{aligned}
P(A = a \mid C = c, D = d) &= \int_{\mathcal{B}} P(A = a \mid B = b, C = c, D = d) dP_B(b \mid C = c, D = d) \\
&= \int_{\mathcal{B}} P(A = a \mid B = b, C = c) dP_B(b \mid C = c, D = d),
\end{aligned}
$$

and the final integrand belongs to $\beta[e^{-\alpha}, e^{\alpha}]$. Combining the preceding three displayed expressions, we find that

$$
\begin{aligned}
|P(A = a \mid B = b, C) - \mathbb{P}(A = a \mid C)| &\leq \left(e^{\alpha} - e^{-\alpha}\right) \beta \\
&\leq \left(e^{\alpha} - e^{-\alpha}\right) e^{\alpha} \min \left\{P(A = a \mid C), P(A = a \mid C, D)\right\}.
\end{aligned}
$$

This completes the proof of the upper bound (46).

It remains to prove inequality (48). We observe from expression (47) that

$$
P(A = a \mid B = b, C) = \frac{P(A = a) \Psi_1(a, C)}{\sum_{a' \in \mathcal{A}} P(A = a') \Psi_1(a', C) \frac{p_B(b \mid A = a')}{p_B(b \mid A = a)}}.
$$

By the likelihood ratio bound (45), we have $p_B(b \mid A = a')/p_B(b \mid A = a) \in [e^{-\alpha}, e^{\alpha}]$, and combining this inequality with the above equation yields inequality (48). ∎

## H   Proof of Lemma 1

For any $\Delta > 0$ and any estimator $\widehat{\theta}$, if $V$ is a random variable uniformly chosen from $\mathcal{V}$, then we have

$$
\max_{\nu \in \mathcal{V}} \mathbb{E}\left[\|\widehat{\theta} - \theta_{\nu}\|_2^2\right] \geq \mathbb{E}\left[\|\widehat{\theta} - \theta_V\|_2^2\right] \geq \mathbb{E}\left[\Delta^2 1_{\left(\|\widehat{\theta} - \theta_V\|_2 \geq \Delta\right)}\right] = \Delta^2 \mathbb{P}(\|\widehat{\theta} - \theta_V\|_2 \geq \Delta). \quad (49)
$$

We now lower bound $\mathbb{P}(\|\widehat{\theta} - \theta_V\|_2 \geq \Delta)$ by a testing-like probability claimed in the lemma. Define the testing function

$$
\widehat{\nu} := \underset{\nu \in \mathcal{V}}{\operatorname{argmin}} \|\theta_{\nu} - \widehat{\theta}\|_2.
$$

The triangle inequality implies that

$$
\|\theta_{\widehat{\nu}} - \theta_V\|_2 \leq \|\theta_{\widehat{\nu}} - \widehat{\theta}\|_2 + \|\widehat{\theta} - \theta_V\|_2 \leq 2\|\widehat{\theta} - \theta_V\|_2 \quad (50)
$$

Recall that $\theta_{\nu} = \delta \nu$ where $\nu \in \{-1, 1\}^d$, we have $\|\theta_{\widehat{\nu}} - \theta_V\|_2 = 2\delta \sqrt{d_{\text{ham}}(\widehat{\nu}, V)}$. Combining this equation with inequality (50) implies that

$$
\text{if } d_{\text{ham}}(\widehat{\nu}, V) > t \text{ then } \|\widehat{\theta} - \theta_V\|_2^2 \geq \delta^2(\lfloor t \rfloor + 1).
$$

Consequently,

$$
P\left(\|\widehat{\theta} - \theta_V\|_2^2 \geq \delta^2(\lfloor t \rfloor + 1)\right) \geq P(d_{\text{ham}}(\widehat{\nu}, V) > t). \quad (51)
$$

Combining inequality (49) and (51) with $\Delta^2 = \delta^2(\lfloor t \rfloor + 1)$, we have

$$
\max_{\nu \in \mathcal{V}} \mathbb{E}\left[\|\widehat{\theta} - \theta_{\nu}\|_2^2\right] \geq \delta^2(\lfloor t \rfloor + 1) P(d_{\text{ham}}(\widehat{\nu}, V) > t).
$$

On the righthand side of the above inequality, taking infinium over all testing functions establishes the result.