[Reviews · NeurIPS 2013]

Submitted by Assigned_Reviewer_4

This paper gives minimax lower bound for the number of bits, B, needed to be communicated between m estimators, each of which has acessess to n iid samples, such that the estimation error is on par with the centralized case (with mn samples). This is a very interesting problem of theoretic and practical significance. In view of the fact that the minimax rate without communication constraints which does not admit a general solution, I do not expect a general solution for the minimal B. As the authors shows the conclusion seems problem-specific and can be quite pessimistic, meaning a large amount of communication overhead is necessary. While the proof of Thm 2 is quite trivial (a vanilla application of Fano's inequality), the proof of Thm 3 is quite interesting which relies on a type of strong data processing inequality for mutual information under some bounded density condition, which is reminiscient of differential privacy and Thm 1 in

J. C. Duchi, M. I. Jordan and M. J. Wainwright (2013). Local privacy and statistical minimax rates. Arxiv technical report, February 2013.


- It will be very interested to look at infinite-dim problems (e.g. function estimation under white Gaussian noise) especially those where the parameter set has faster growth such that Yang-Barron gives sharp rate w.r.t. KL loss. Whether one need even bigger B in order to reach the rate at sample size=mn

- I wonder if there is any problem in multi-user information theory that is similar in spirit, where a joint decoder has access to rate-constrained coded messages and want to recover the original source with some fidelity. I am not familiar with this field, but it might be interested to identify this connection. Of course, the major difference here is that there is no large blocklength to exploit.

- The following paper might be relevant (the reviewer has NOT read the paper)

Maxim Raginsky, Achievability results for statistical learning under communication constraints
Proceedings of ISIT 2009

- "this work does not characterize the effects of limiting communication" -> "these results do not characterize the effects of limited communication"

- The well-known statistical minimax rate for linear regression scales as d \sigma^2 /(\lambda 2 nm) (e.g. [15]). -- I am not so sure about this statement. The result of [15] is known to be loose in finite-dim models. Also what conditions on the design matrices A are assumed?
Summary: This is a well-written and interesting paper. A clear accept.

Submitted by Assigned_Reviewer_7

This paper considers minimax lower bounds on the mean squared error for statistical estimation problems, in a setting where available data is distributed i.i.d. across several machines and there is a constraint on the number of bits that may be communicated. The paper considers both interactive and non-interactive protocols, where the final estimate is a function of the messages sent by the machines. For interactive protocols, these include lower bounds for estimating the mean of a uniform distribution in a unit cube, and the mean of a univariate Gaussian distribution. For non-interactive protocols, the lower bounds are for estimating the mean of a multivariate Gaussian distribution, and applications for linear and probit regression in a fixed (known) design setting.

The issue of distributed learning and estimation has received plenty of attention in recent years, mostly in the form of algorithms and upper bounds. This paper is among the first (see remarks below) to explicitly consider lower bounds. Since lower bounds are crucial to understanding the limits of achievable performance, I consider this a very interesting line of research. The results presented here seem to be interesting first steps in this direction. Also, I found the paper clear and quite easy to follow. Finally, the results seem overall correct, although I haven't checked the proofs in the supplementary carefully.

My main criticism is that the results focus on somewhat limited regimes, which may not correspond to realistic applications of distributed learning. To be more specific:

1) The setting considered here is only harder than the standard (non-distributed) setting of statistical estimation. Thus, all the standard (non-distributed) lower bounds trivially hold. The lower bounds presented here are stronger than those standard bounds, only in the regime where the average amount of communication B_i per machine is smaller than the data dimension d. However, this is a very harsh constraint. For example, in standard linear prediction, it means that each machine can't even output a single predictor. For most applications of distributed learning, one is very happy with protocols where B_i is order of d, and for such a regime, the results here do not improve on the existing standard lower bounds. For fairness, it should be said that it's not known whether the standard lower bounds are improvable when B_i >= d, except in some cases, such as statistical estimation with sufficiently smooth loss functions (see Zhang et al. 2012 - reference [18] in the paper). These results do not apply to non-smooth losses, e.g. when we try to distribute support vector machines. Moreover, all these results are in terms of mean-squared-error, and do not give tight results when converting them to optimization error / excess risk (e.g. in terms of dependence on the strong-convexity parameter, which can be small in regularized statistical learning problems). Anyway, the results in this paper do not shed light on these issues.

2) Most of the results here are for non-interactive protocols, where each machine can only send a single message based on its data, and the final estimator is a function of these messages. This is restrictive, since one can easily imagine algorithms which require several communication rounds between the machines. The concrete results for interactive protocols (proposition 1 and theorem 2) seem much more limited, focusing either on a uniform distribution in a unit cube, or a 1-dimensional setting. As to the more generic theorem 1, I may be wrong, but I think the same proof would apply to a non-distributed setting, where the only constraint it that the output \hat{\theta} is limited to B bits. So in a certain sense, it may not capture an interesting aspect of the distributed setting.

3) The regression lower bounds focus on a fixed design setting, where the design/instances are known by all machines in advance, and the uncertainty is in the target values. In a distributed setting, it's much more realistic to assume unknown instances partitioned between the machines.



Technical Comments
------------------

- Line 41: "Currently there is little theoretical understanding of the role of communication complexity in statistical inference, and thus the design
and analysis of distributed inference algorithms is more art than science". I think this is exaggerated. Principled approaches to design and analyze distributed inference and learning have been well-studied in the literature. It is true that lower bounds have not been as carefully studied.

- It would be good to explain why the dependence on B is so different between the two settings of proposition 1 and theorem 2.

- Line 111: "an protocol"

- Line 119, definition of minimax risk - why is the protocol Pi separated from the estimator function \Theta? Pehaps it would make more sense to include the final estimate computation as part of the protocol.

- Line 148: If I understood correctly, example 1 shows the tightness of Theorem 1 only for one particular choice of B. It would be better if one can come up with an example which shows the tightness of the bound for *any* choice of B.

- Line 1120 in supplementary: "achieves minimax" -> "which achieves the"(?)

- In discussing related work, the authors should also mention the COLT 2012 paper "Distributed Learning, Communication Complexity and Privacy" by Balcan et al., which also discuss lower bounds for distributed learning (although in a different setting).

- Section 2: When defining A_ind and A_inter, do any of the lower bounds become better if we require L<=B always rather just in expectation? Assuming that not, it might be worthwhile to point this out.
Summary: The paper provides lower bounds for statistical estimation in a data-distributed setting. These are interesting first steps in this direction, although the regimes considered are somewhat limited.

Submitted by Assigned_Reviewer_9

This paper is well-described by its title - it gives minimax lower bounds (and some matching upper bounds) for statistical estimation problems when data is split across several machines given a constraint on how much communication is allowed (independently or interactively) between the servers having the data and a fusion point where the messages from the servers are collected and a final estimate is reached. This is done in a variety of settings, all of which were followable, and the lower bounds are proved using standard reductions to hypothesis testing followed by some non-standard detailed calculations (that I did not completely follow).

I think the paper was well-structured and easy to read, going from more examples and intuition to proofs at the end, with good intuition for the start of the proof. To the best of my knowledge, the work was original, it is definitely of significance to the community, and it is of high quality.

I think more emphasis should be placed on the differences in rates between interactive and independent settings. Also, the proof on page 8 has descriptions for each step, but I feel better intuition can be given (in section 5.2, in addition to section 5.1).
Summary: Overall, I think this is a very good paper and highly recommend its acceptance - it was clear, high quality, original and very significant.
Author Feedback

Author rebuttal: We thank the reviewers for their careful reading and comments and criticism of
our paper. We will address all comments in the revised version of the paper;
we tackle major feedback below.

REVIEWER 4

We appreciate the positive feedback, and indeed, Thm. 3 is reminiscent of
Duchi et al.'s work--we will provide a citation. There is some difference
because our likelihood ratio inequalities (in the case of Theorems 2 and 4,
e.g. Lemma 3 and Eq. (31) in the appendix) do not hold for all samples, and
the data processing inequalities (at least for now) require somewhat careful
conditions on the random variable V. But certainly they do have substantial
similarities; we are quite curious to understand when such likelihood
ratio/bounded density conditions give data processing inequalities.

- The infinite dimensional question is extremely interesting--we will be
looking at that in the future (we have seen some recent work on communication
upper bounds here; see Zhang, Duchi, Wainwright, COLT 2013).

- The connections with multi-user information theory are interesting; we do
not have a full understanding yet. Slepian-Wolf coding is the closest problem
in spirit to ours; here senders encode independently and a decoder decodes
independently encoded messages--it is possible in this setting (since large
blocklengths can be exploited) to send messages at a rate of the joint
entropy; in the statistical estimation setting it appears we need to send a
bit more.

- In terms of linear regression bounds, we are assuming that A has bounded
minimum and maximum singular values--we will be more precise. (The
true minimax rate is tr(((1/n) A'*A)^{-1}) / n).

Thanks also for the pointer to the Raginsky paper. The settings are slightly
different--Raginsky considers a single communication channel--but it is good
to have the reference.

REVIEWER 7

1) The reviewer is correct--our setting is only harder. But given the
difficulty--and, to our knowledge, lack of lower bounds on communication (and
the fact that sending O(d) bits is sufficient for some problems)--of providing
such lower bounds, we think this is a good step in this direction. Multi-user
information theory, even after fifty years of work, is still a quite unsolved
field.

2) Theorem 1 is a straightforward application of Fano's inequality, so yes, it
also applies in non-distributed settings. Our main contributions are Theorems
2-4, which *are* distributed. But Theorem 1 is sharp for some settings, as
exemplified by Proposition 1, where estimating a uniform family can be done
with exponentially fewer (in m) bits with interactivity than without
(or for the normal distribution).

3) We provide *lower* bounds for the *easier* fixed-design regression problem;
a harder problem (where the design matrices are unknown) will have worse
lower bounds. Given Zhang et al.'s [18] results, however, our lower bounds
are sharp in this case as well.

Technical comments:

- We view an understanding of communication complexity to be fleshed out
only when there are demonstrable tradeoffs, which necessarily include lower
bounds.

- Dependence on B: The compactness of support of the uniform appears
to allow improvements. We will explain this more.

- 119: The protocol and \theta are both infimized over; they may interact
and our definition does not exclude this. We will be clearer in the final
version.

- 148: The bound is tight for essentially any 1-machine setting; we may either
choose the number of samples n or the communication bound B to make it tight.

- We will discuss connections to Balcan et al. Given the difference in
settings, the connections are not immediate.

- Section 2: The lower bounds become somewhat easier to prove when L \le B,
but Theorems 2-4 do not change. The upper bound in Proposition 1 requires
randomized communication; we believe a \log(m) dependence in the upper bound
is impossible without randomization. (Each processor must communicate at least
1 bit in a non-random setting.)

REVIEWER 9

We think the title is pretty descriptive as well :).

Does the reviewer have suggestions for parts of Section 5 that are unclear?
We would certainly like the intuition to come through clearly and would
appreciate any suggestions. We recognize that after the standard lower bound
(through Eq. (12), essentially), the calculations become quite complex and a
bit non-intuitive. Perhaps making clearer the importance of the strong
data-processing inequality (15), which is the essential part of the argument,
and its connections with the probability of error in identifying the V_j
would be helpful.